# Approximate Allocation Matching for Structural Causal Bandits with Unobserved Confounders

**Lai Wei**
Life Sciences Institute
University of Michigan
weilatim@gmail.com

**Muhammad Qasim Elahi**
Electrical and Computer Engineering
Purdue University
elahi0@purdue.edu

**Mahsa Ghasemi**
Electrical and Computer Engineering
Purdue University
mahsa@purdue.edu

**Murat Kocaoglu**
Electrical and Computer Engineering
Purdue University
mkocaoglu@purdue.edu

## Abstract

Structural causal bandit provides a framework for decision-making problems when causal information is available. It models the stochastic environment with a structural causal model (SCM) that governs the causal relations between random variables. In each round, an agent applies an intervention (or no intervention) by setting certain variables to some constants, and receives a stochastic reward from a non-manipulable variable. Though the causal structure is given, the observational and interventional distributions of these random variables are unknown beforehand, and they can only be learned through interactions with the environment. Therefore, to maximize the expected cumulative reward, it is critical to balance the exploration-versus-exploitation tradeoff. We consider discrete random variables with a finite domain and a semi-Markovian setting, where random variables are affected by unobserved confounders. Using the canonical SCM formulation to discretize the domains of unobserved variables, we efficiently integrate samples to reduce model uncertainty, gaining an advantage over those in a classical multi-armed bandit setup. We provide a logarithmic asymptotic regret lower bound for the structural causal bandit problem. Inspired by the lower bound, we design an algorithm that can utilize the causal structure to accelerate the learning process and take informative and rewarding interventions. We establish that our algorithm achieves a logarithmic regret and demonstrate that it outperforms the existing methods via simulations.

## 1 Introduction

Sequential decision-making in uncertain environments is one of the most fundamental problems across scientific disciplines, including robotics, economics, social science, clinical trials, and agriculture [32, 26, 36, 7, 3]. In almost all these applications, it is required to implement interventions to control some aspects of the system to optimize an outcome of interest. In economics and political science, the government could use fiscal policy including changing the interest rate and taxation to achieve regulation and control of a country's economy. In this paper, we deal with designing an adaptive policy in an uncertain stochastic environment to learn and apply optimal interventions.

In sequential decision-making problems, it is critical to balance the explore-versus-exploit tradeoff, which deals with choosing between informative options and empirically more rewarding alternative options. The multi-armed bandit (MAB) [33, 30, 18] is a classical problem formulation for achieving such a tradeoff. In MAB, a decision-maker sequentially chooses an action corresponding to pulling

37th Conference on Neural Information Processing Systems (NeurIPS 2023).

an arm from a set of arms, each generating stochastic rewards with probability distribution unknown to the decision-maker. The objective is to achieve the maximum cumulative reward over a time horizon, or equivalently to minimize the regret, which measures suboptimality in cumulative rewards against selecting the best action all the time. The classic stochastic bandit formulation only considers the connection between an action and its expected reward. However, in real-world applications, an action can affect the entire system in a more complex way. As the agricultural example given in [19], changing the moisture level could affect the temperature and nutrient absorption, which are also crucial to crop yields. The relations between such factors cannot be characterized by the MAB.

Causal knowledge enables us to model the causal relations between multiple factors in the environment. In machine learning research, it has been noted that causal models could help improve the generalization performance when data distribution shifts [11], and address unstable translation of successful reinforcement learning methods in simulation to real-world problems [12] such as self-driving cars and recommendation systems. The casual bandit models the stochastic environment with a causal graph that connects a set of random variables of interest, and actions are modeled by interventions on different subsets of nodes. It provides a critical interpretation of causal influences among random variables [28] that allows us to reason about what will happen to the environment after certain interventions are made to the data-generating process. Compared to the classic MAB with only reward feedback, the casual bandit represents a more detailed model of real-world decision-making problems, including richer feedback from multiple random variables and the causal relations among them. By applying causal reasoning, the learning process can be accelerated and more rewarding interventions can be selected. In applications where the space of intervention is large and explicit experiments are costly, such as social science or economics, it has special value.

Due to the reasons mentioned above, there has been a recent surge of interest in the research community on causal decision-making. Some early attempts include [6], in which the authors took a causal approach to study how to make final decisions based on natural predilections, and proposed an algorithm by modifying Thompson sampling [33]. In [19], the investigators started from a parallel causal structure and extended the algorithm to the general causal bandit problem. Subsequent work [37] took preprocessing steps to get rough estimates of interventional distributions and use them to design efficient experiments to discover the best intervention. In [1], the Bayesian optimization technique is employed to solve the same problem.

The most closely related works focusing on regret minimization in the structural causal bandit with confounders are [22, 23]. In both works, the authors took a graphical characterization using do-calculus [28] to reduce the large intervention set to the possibly optimal minimal intervention set and run a KL-UCB algorithm [14] within the reduced action space. The later work is an extension of the previous one that includes non-manipulable variables. Other lines of causal bandit research normally assume additional prior information, such as infinite observational data [39] or marginal interventional distributions [24, 8]. A general linear model is adopted in [13], which is different from the non-parametric setup in this paper.

In this paper, we focus on the MAB sequential decision-making problem where the arms correspond to interventions on an arbitrary causal graph, including unobserved confounders affecting multiple pairs of variables (also known in causal terminology as the semi-Markovian setting). We recognize the structural causal bandit falls into the category of structured bandit [10, 35]. It has been noted the classical upper confidence bound (UCB) algorithm and Thompson sampling may not fully leverage structural information [31, 20]. It is worth considering the potential limitations of algorithms based on these ideas, such as CRM-AL [25] and generalized Thompson sampling [27]. To address this issue and make systematic use of a known causal structure, we design an algorithm that can meticulously utilize causal structural information. In particular, we make the following contributions:

- We provide a logarithmic asymptotic regret lower bound for the structural causal bandit problem with latent confounders by leveraging the canonical SCM formulation [40] to create a space of all possible interventional distributions given a causal graph.

- Inspired by the lower bound, we extend [22] and apply approximate allocation matching to design an algorithm that can effectively utilize causal information provided by the causal graph to accelerate the learning process and take informative and rewarding interventions.

- We analyze the algorithm to provide a problem-dependent logarithmic upper bound on expected regret and compare it with [22] to show it has a better theoretical guarantee.

- We complement the theoretical results by evaluating the proposed approach in a variety of experimental settings featuring different causal structures. We show that our algorithm outperforms the existing baselines in terms of empirical regret.

## 2 Problem Statement and Background

In this section, we present the formulation of the structural causal bandit problem with latent confounders. We follow the convention to represent a random variable and its value in a capital letter and a lowercase letter respectively. A multivariate random variable is represented in a bold letter.

### 2.1 Structural Causal Model

Our approach adopts the structural causal model (SCM) [29] to provide a causal perspective on the data-generating process. This allows us to express the relationship between variables and capture important causal concepts, including unobserved variables, observational distributions, and interventional distributions. For a random variable $\mathbf{X}$, let $\Omega(\mathbf{X})$ denote its domain.

**Definition 1** (SCM). *A structural causal model is a 4-tuple $\langle \mathbf{U}, \mathbf{V}, \mathbb{F}, P(\mathbf{U}) \rangle$ where:*

1. $\mathbf{U}$ *is a set of unobserved independent background variables (also called exogenous), that determine the randomness of the model.*

2. $\mathbf{V} := \{V_i \mid i = 1, \ldots, n\}$ *is the set of observable variables (also called endogenous).*

3. $\mathbb{F}$ *is a set of structural equations $\{f_1, f_2, ..., f_n\}$ such that for each $V_i \in \mathbf{V}$, $f_i : \Omega(\mathbf{U}_i) \times \Omega(\mathsf{Pa}_i) \to \Omega(V_i)$ defines a mapping, where $\mathbf{U}_i \subseteq \mathbf{U}$ and $\mathsf{Pa}_i \subseteq \mathbf{V}$ is the parent set of $V_i$. So $\mathbb{F}$ as a whole determines a mapping relationship from $\Omega(\mathbf{U})$ to $\Omega(\mathbf{V})$.*

4. $P(\mathbf{U})$ *is a joint probability distribution over all the exogenous variables.*

Each structural causal model is associated with a causal graph $\mathcal{G} = \langle \mathbf{V}, \mathcal{E}, \mathcal{B} \rangle$, where the node set $\mathbf{V}$ corresponds to the set of observable variables, $\mathcal{E}$ is the set of directed edges and $\mathcal{B}$ is the set of bidirected edges. For each $V_j \in \mathsf{Pa}_i$, there exists a directed edge in $\mathcal{E}$ such that $V_j \to V_i$ which indicates functional dependency from $V_j$ to $V_i$. Without showing unobserved exogenous variable $\mathbf{U}$ explicitly, the confounding effects of $\mathbf{U}$ are represented using bidirected edges in $\mathcal{B}$, which connect multiple pairs of observable variables in $\mathbf{V}$. The presence of $V_i \leftrightarrow V_j$ in $\mathcal{G}$ represents unmeasured factors (or confounders) that may influence both $V_i$ and $V_j$. In other words, $f_i$ and $f_j$ share common exogenous variables $\mathbf{U}_i \cap \mathbf{U}_j \neq \emptyset$ as input. Let $|\Omega(\mathbf{V})|$ be the cardinality of $\Omega(\mathbf{V})$.

**Assumption 1.** *We assume each $V \in \mathbf{V}$ can only take a finite number of values, i.e., $|\Omega(\mathbf{V})|$ is finite.*

Within the causal graph, an intervention on a subset of random variables $\mathbf{S} \subseteq \mathbf{V}$ denoted by the do-operator $do(\mathbf{S} = \mathbf{s})$ sets the structural equation for each $S_j \in \mathbf{S}$ to be $S_j = s_j$. We also refer to $\mathbf{s}$ as intervention for brevity. If a node $V_i \notin \mathbf{S}$, its structural equation remains to be $V_i = f_i(\mathbf{U}_i, \mathsf{Pa}_i)$. The empty intervention denoted by $do(\emptyset)$ does not change the structural equation of any random variable, and the distribution of $\mathbf{V}$ is also called observational distribution. Let $\mathbb{1}$ denote the indicator function. An SCM induces interventional distributions: if $\mathbf{v}$ is consistent with intervention $\mathbf{s}$,

$$P(\mathbf{v} \mid do(\mathbf{S} = \mathbf{s})) = \sum_{\mathbf{u}} P(\mathbf{u}) \prod_{V_i \notin \mathbf{S}} \mathbb{1}\{v_i = f_i(\mathsf{pa}_i, \mathbf{u}_i)\} := P_{\mathbf{s}}(\mathbf{v}). \tag{1}$$

Otherwise, $P(\mathbf{v} \mid do(\mathbf{S} = \mathbf{s})) = 0$. For a more detailed discussion on SCMs, we refer to [29].

### 2.2 Causal Bandit Problem with Confounders

The causal bandit problem with confounders studies the sequential decision-making problem with causal information provided by a causal graph $\mathcal{G} = \langle \mathbf{V}, \mathcal{E}, \mathcal{B} \rangle$ with confounding effects represented by bidirected edge set $\mathcal{B}$. The causal graph $\mathcal{G}$ is given but the exact interventional and observational distributions are unknown. Without loss of generality, we assume the reward is collected from node $V_n$, which is bounded in $[0, 1]$ and can not be intervened on. Let $\mathcal{I} \subseteq \{\mathbf{s} \in \Omega(\mathbf{S}) \mid \mathbf{S} \subseteq \mathbf{V} \setminus V_n\}$ be the space of allowed interventions. The decision-maker is required to decide which intervention to apply

at each time step. The expected reward of intervention $\mathbf{s} \in \mathcal{I}$ is $\mu_{\mathbf{s}} = \mathbb{E}_{P_{\mathbf{s}}}[V_n] = \sum_{\mathbf{v} \in \Omega(\mathbf{V})} v_n P_{\mathbf{s}}(\mathbf{v})$, where $P_{\mathbf{s}}$ is defined in (1) and $\mathbb{E}_P$ means the expectation is computed with probability distribution $P$.

At each time $t \in \{1, \ldots, T\}$, the agent follows an adaptive allocation policy $\pi$ to select an intervention $\mathbf{s}_t \in \mathcal{I}$ and observe the causal bandit feedback $\mathbf{V}_t \sim P_{\mathbf{s}_t}(\mathbf{V})$, which is a realization of all observable variables after intervention. In fact, policy $\pi$ is a sequence $\{\pi_t\}_{t \in \mathbb{N}}$, where each $\pi_t$ determines the probability distribution of taking intervention $\mathbf{S}_t \in \mathcal{I}$ given intervention and observation history $\pi_t(\mathbf{S}_t \mid \mathbf{s}_1, \mathbf{v}_1, \ldots, \mathbf{s}_{t-1}, \mathbf{v}_{t-1})$. Let $\mu_* := \max_{\mathbf{s} \in \mathcal{I}} \mu_{\mathbf{s}}$ and $\Delta_{\mathbf{s}} = \mu_* - \mu_{\mathbf{s}}$ denote the optimal mean reward and the expected optimality gap of intervention $do(\mathbf{S} = \mathbf{s})$ respectively. Given a causal graph $\mathcal{G}$, the objective of the causal bandit problem is to design a policy $\pi$ to maximize the expected cumulative reward, or equivalently, to minimize the *expected regret*:

$$R_T^{\pi} := \mathbb{E}\left[ \sum_{t=1}^{T} (\mu_* - V_{n,t}) \right] = \sum_{\mathbf{s} \in \mathcal{I}} \mathbb{E}[N_T(\mathbf{s})] \Delta_{\mathbf{s}}, \tag{2}$$

where $N_T(\mathbf{s})$ is the total number of times intervention $do(\mathbf{S} = \mathbf{s})$ is taken by policy $\pi$ until time horizon till $T$, and the expectation is computed over different realizations of $\{\mathbf{S}_t, \mathbf{V}_t\}_{t=1}^{T}$ from the interactions between the random policy $\pi$ and the causal bandit model $\langle P_{\mathbf{s}} \rangle_{\mathbf{s} \in \mathcal{I}}$. Thus, $R_T^{\pi}$ is the gap between the expected cumulative rewards of selecting the best intervention all the time and that by the policy $\pi$. The regret decomposition in (2) is from [18], and it expresses the expected regret with products of the expected number of suboptimal action selections and expected reward gaps.

The large action space in causal bandits poses many challenges to the learning problem. It is desirable to reduce the action space without excluding all optimal interventions. The do-calculus introduced in [28] provides a set of guidelines for evaluating invariances across interventions. In this context, our specific focus is on Rule 3, which provides the conditions under which a series of interventions have no impact on the outcome variable. For example, Rule 3 implies that $P_{\mathbf{W}, \mathbf{Z}}(y) = P_{\mathbf{Z}}(y)$ if we have $(\mathbf{Y} \perp\!\!\!\perp \mathbf{W} | \mathbf{Z})_{\mathcal{G}_{\overline{\mathbf{W} \cup \mathbf{Z}}}}$, where $\mathcal{G}_{\overline{\mathbf{W} \cup \mathbf{Z}}}$ is the subgraph of DAG $\mathcal{G}$ with incoming edges to the set $\mathbf{W} \cup \mathbf{Z}$ removed. This concept leads to the notion of a minimal intervention set (MIS), which is a subset of variables $\mathbf{S} \subseteq \mathbf{V} \setminus \mathbf{V_n}$ such that there is no $\mathbf{S}' \subset \mathbf{S}$ for which $\mu_{\mathbf{s}[\mathbf{S}']} = \mu_{\mathbf{s}}$ holds for every Structural Causal Model (SCM) with a causal graph $\mathcal{G}$ [22]. Here, $\mathbf{s}[\mathbf{S}']$ denotes an intervention on $\mathbf{S}' \cap \mathbf{S}$ with values consistent with $\mathbf{s}$ . An MIS $\mathbf{S}$ is considered a possibly optimal MIS (POMIS) if some intervention $\mathbf{s} \in \Omega(\mathbf{S})$ can achieve the optimal mean reward in an SCM with a causal graph $\mathcal{G}$. We define $\mathcal{I} = \{\mathbf{s} \in \Omega(\mathbf{S}) \mid \mathbf{S} \text{ is a POMIS}\}$.

At an intuitive level, it may seem logical that the most effective course of action would be to intervene on the immediate causes (parents) of the reward variable $V_n$. This approach would provide a higher level of control over $V_n$ within the system. If the reward variable $V_n$ is not confounded with any of its ancestors, its parent set $\mathsf{Pa}_n$ is the only POMIS. In more general causal graphs where $V_n$ is confounded with any of its ancestors, the paper [22] proves that multiple POMISs exist, which can include variables that are not parents of $V_n$. The paper also provides graphical criteria and an efficient algorithm for constructing a set of all POMISs for a given causal graph. For an effective method to search for $\mathcal{I}$, we refer to [22] and rely on their graphical criteria to construct a set of POMISs. This set of POMISs is used to form the set of possible optimal actions $\mathcal{I}$ for use in the causal bandit algorithm. We make the following assumption, which we expect to hold in a general scenario [38], except for certain specifically designed SCMs.

**Assumption 2.** *Within interventions on POMISs, there exists a unique optimal intervention* $\mathbf{s}_*$.

## 3 Response Variables and Space of Interventional Distribution Tuples

The unspecified domain of $\mathbf{U}$ makes it inconvenient to be directly applied to the learning problem. However, it is noted in [29, Ch. 8] that if each variable in $\mathbf{V}$ takes finite states, $\Omega(\mathbf{U})$ can be partitioned and $\mathbf{U}$ can be projected to a collection of finite-state response variables. The resulting model is equivalent to the original one with respect to all observational and interventional distributions. Such a technique was used to bound the causal and counterfactual effects with observational distribution [29, Ch. 8]. We take it to define a parameterized space of interventional distribution tuples.

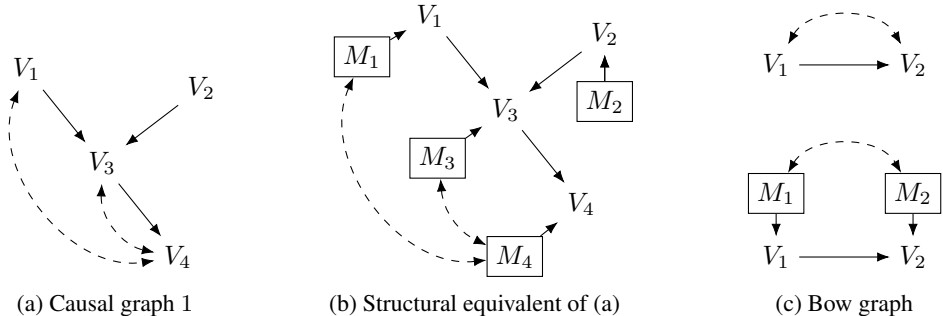

(a) Causal graph 1        (b) Structural equivalent of (a)        (c) Bow graph

Figure 1: Causal graphs and their structural equivalents

## 3.1 Response Variables and Canonical Structural Causal Model

In Definition 1, each structural equation $V_i = f_i(\mathbf{U}_i, \mathsf{Pa}_i)$ connects random variable $V_i$ to its parents $\mathsf{Pa}_i$. As $\mathbf{U}_i$ varies along its domain, regardless of how complex the variation is, the only effect it can have is to switch the relationship between $\mathsf{Pa}_i$ and $V_i$. Since there are at most $|\Omega(V_i)|^{|\Omega(\mathsf{Pa}_i)|}$ mapping relationships from $\mathsf{Pa}_i$ to $V_i$, we can decompose $f_i$ as follows,

$$V_i = f_i(\mathbf{U}_i, \mathsf{Pa}_i) = \underline{f}_i(M_i, \mathsf{Pa}_i), \ M_i = \overline{f}_i(\mathbf{U}_i), \tag{3}$$

where $M_i \in \left\{0, \ldots, |\Omega(V_i)|^{|\Omega(\mathsf{Pa}_i)|} - 1\right\}$ is a random variable corresponds to a mapping from $\Omega(\mathsf{Pa}_i)$ to $\Omega(V_i)$. For an observable variable without parents, we simply have $M_i \in \{0, \ldots, |\Omega(V_i)| - 1\}$. Such a random variable $M_i \in \mathbf{M}$ attached to each $V_i \in \mathbf{V}$ is called response variable in [5]. Treating response variables as exogenous variables, the resulting SCM $\langle \mathbf{M}, \mathbf{V}, \underline{\mathbb{F}}, P(\mathbf{M}) \rangle$ is called *canonical SCM* in [40], where $\underline{\mathbb{F}} = \{\underline{f}_1, \underline{f}_2, ..., \underline{f}_n\}$.

The decomposition of structural equations in equation (3) can be represented graphically by including response variables in $\mathbf{M}$. Since $V_i = \underline{f}_i(M_i, \mathsf{Pa}_i)$, for each $V_i$ there is a directed edge $M_i \to V_i$. We also know that if there exists $V_i \leftrightarrow V_j$ in $\mathcal{G}$, both $V_i$ and $V_j$ are affected by common exogenous variables $\mathbf{U}_i \cap \mathbf{U}_j \neq \emptyset$. As a result, $M_i$ and $M_j$ are correlated and they are also connected by a bidirected edge. For example, the causal graph in Fig. 1(a) has a structural equivalent in Fig. 1(b).

Given intervention $\mathbf{s} \in \mathcal{I}$, there exists a deterministic relationship from $\mathbf{M}$ to $\mathbf{V}$. To further explain the idea, consider the bow graph example in Fig. 1(c), where $V_1, V_2$ are binary variables taking value 0 or 1. As illustrated above, the response variables are $M_1 \in \{0, 1\}$ and $M_2 \in \{0, 1, 2, 3\}$, and the structural equations for $V_1$ and $V_2$ are as follows:

$$V_1 = \underline{f}_1(M_1) = M_1, \ V_2 = \underline{f}_2(M_2, V_1) = \begin{cases} 0 & M_2 = 0 \\ V_1 & M_2 = 1 \\ 1 - V_1 & M_2 = 2 \\ 1 & M_2 = 3 \end{cases}.$$

If $do(\emptyset)$ is applied, $(V_1, V_2) = (0, 1)$ when $(M_1, M_2) = (0, 2)$ or $(0, 3)$. If $do(V_1 = 1)$ is applied, $(V_1, V_2) = (1, 0)$ when $M_1 = 0$ or 1 and $M_2 = 0$ or 2. It shows that for a given intervention $do(\emptyset)$ or $do(V_1 = 1)$, there exists a deterministic mapping from $\mathbf{M}$ to $\mathbf{V}$. Besides, the true value of $\mathbf{M}$ can only be inferred from $\mathbf{V}$ since $\mathbf{M}$ is unobservable. Since $M_1$ and $M_2$ are correlated, we define their joint distribution as $P(M_1 = i, M_2 = j) = p_{ij}$. We can further express the observational and interventional probabilities of $\mathbf{V}$ with the probability of $\mathbf{M}$ as follows,

$$
\begin{aligned}
P(V_1 = 0, V_2 = 0) &= p_{00} + p_{01}, & P(V_1 = 0, V_2 = 1) &= p_{02} + p_{03}, \\
P(V_1 = 1, V_2 = 0) &= p_{10} + p_{12}, & P(V_1 = 1, V_2 = 1) &= p_{11} + p_{13}, \\
P_{do(V_1=0)}(V_2 = 0) &= p_{00} + p_{01} + p_{10} + p_{11}, & P_{do(V_1=0)}(V_2 = 1) &= p_{02} + p_{03} + p_{12} + p_{13}, \\
P_{do(V_1=1)}(V_2 = 0) &= p_{00} + p_{02} + p_{10} + p_{12}, & P_{do(V_1=1)}(V_2 = 1) &= p_{01} + p_{03} + p_{11} + p_{13}.
\end{aligned}
\tag{4}
$$

It can be seen these parameters are sufficient for specifying the model of Fig. 1(c). Such a parameterization technique can be extended to general causal graphs with latent confounders [40].

## 3.2 Space of Interventional Distribution Tuples

In a causal graph, a confounded component (c-component) is a maximal set of vertices connected with bidirected edges [34]. Note that a singleton node without bidirected edges is also a c-component. For example, there are two c-components in Fig. 1(a): $\{V_1, V_3, V_4\}$ and $\{V_2\}$. In fact for any causal graph $\mathcal{G}$, its observable variables $\mathbf{V}$ can be uniquely partitioned into c-components $\{\mathbf{V}^1, \ldots, \mathbf{V}^{n_c(\mathcal{G})}\}$, where $n_c(\mathcal{G})$ is the total number of c-components in $\mathcal{G}$. Then $\mathbf{M}$ can also be partitioned into $\{\mathbf{M}^1, \ldots, \mathbf{M}^{n_c(\mathcal{G})}\}$, where each $\mathbf{M}^j$ contains response variables adjacent to $\mathbf{V}^j$. For example in Fig. 1(b), $\mathbf{M}$ is partitioned into $\{M_1, M_3, M_4\}$ and $\{M_2\}$. Within each $\mathbf{M}^j$, the response variables are correlated since they are connected by bidirected edges. Besides, $\mathbf{M}^1, \ldots, \mathbf{M}^{n_c(\mathcal{G})}$ are mutually independent since $\mathbf{M}^i$ and $\mathbf{M}^j$ are not connected with bidirected edges for any $i \neq j$. As a result, $P(\mathbf{m}) = \prod_{j=1}^{n_c(\mathcal{G})} P(\mathbf{m}^j)$. By concatenating $P(\mathbf{m}^j)$ for each $\mathbf{m}^j \in \Omega(\mathbf{M}^j)$, we construct a vector $\mathbf{p}_j \in \Delta(|\Omega(\mathbf{M}^j)|)$, where $\Delta(|\Omega(\mathbf{M}^j)|) := \{\mathbf{p}'_j \in \mathbb{R}_{\geq 0}^{|\Omega(\mathbf{M}^j)|} \mid \mathbf{1}^\top \mathbf{p}'_j = 1\}$ and $\mathbf{1}^\top$ is the transpose of the all-ones vector.

Let the parent set of a c-component $\mathbf{V}^j$ be $\mathsf{Pa}_{\mathbf{V}^j} := (\cup_{i:V_i \in \mathbf{V}^j} \mathsf{Pa}_i) \setminus \mathbf{V}^j$. When taking intervention $do(\mathbf{S} = \mathbf{s})$, the values of $\mathbf{V}^j \cap \mathbf{S}$ is set to $\mathbf{s}[\mathbf{V}^j]$, which denotes the values of $\mathbf{V}^j \cap \mathbf{S}$ that is consistent with $\mathbf{s}$. Notice that $\mathbf{M}^j$ picks the mapping functions from $\mathsf{Pa}_i$ to $V_i$ for all $V_i \in \mathbf{V}^j$. Seeing values $\mathbf{v}^j, \mathsf{pa}_{\mathbf{V}^j}$ and $\mathbf{s}[\mathbf{V}^j]$, there exists a set of configurations of $\mathbf{M}^j$, denoted by $B_{\mathcal{G}, \mathbf{s}[\mathbf{V}^j]}(\mathbf{v}^j, \mathsf{pa}_{\mathbf{V}^j}) \subseteq \Omega(\mathbf{M}^j)$, that can make this happen. By marking configurations in $B_{\mathcal{G}, \mathbf{s}[\mathbf{V}^j]}(\mathbf{v}^j, \mathsf{pa}_{\mathbf{V}^j})$ with 1 and others to be 0, we construct a vector $b_{\mathcal{G}, \mathbf{s}[\mathbf{V}^j]}(\mathbf{v}^j, \mathsf{pa}_{\mathbf{V}^j}) \in \{0, 1\}^{|\Omega(\mathbf{M}^j)|}$. With $\mathbf{M}^1, \ldots, \mathbf{M}^{n_c(\mathcal{G})}$ being mutually independent, the interventional distribution can be factorized as

$$P_{\mathbf{s}}(\mathbf{v}) = \prod_{j=1}^{n_c(\mathcal{G})} P\big(\mathbf{M}^j \in B_{\mathcal{G}, \mathbf{s}[\mathbf{V}^j]}(\mathbf{v}^j, \mathsf{pa}_{\mathbf{V}^j})\big) = \prod_{j=1}^{n_c(\mathcal{G})} b_{\mathcal{G}, \mathbf{s}[\mathbf{V}^j]}^\top(\mathbf{v}^j, \mathsf{pa}_{\mathbf{V}^j}) \mathbf{p}_j. \tag{5}$$

In the bow graph example in Fig. 1(c), there is one c-component, and vector $b_{\mathcal{G}, \mathbf{s}[\mathbf{V}^j]}^\top(\mathbf{v}^j, \mathsf{pa}_{\mathbf{V}^j})$ can be constructed by referring to (4), where $\mathbf{p}_1$ is a concatenation of $p_{i,j}$. The result in (5) generalizes the parameterization for the bow graph to a general causal graph. Based on it, given a causal graph $\mathcal{G}$, we define the space of interventional distribution tuples as the following.

**Definition 2** (Space of Interventional Distribution Tuples). *Given a causal graph $\mathcal{G} = \langle \mathbf{V}, \mathcal{E}, \mathcal{B} \rangle$, the space of interventional (and observational) distribution tuples $\mathcal{P} = \langle P_{\mathbf{s}} \rangle_{\mathbf{s} \in \mathcal{I}}$ is*

$$\mathcal{M}_{\mathcal{G}} := \left\{ \mathcal{P}' \;\middle|\; \forall \mathbf{s} \in \mathcal{I}, \mathbf{v} \in \Omega(\mathbf{V}) : P'_{\mathbf{s}}(\mathbf{v}) = \prod_{j=1}^{n_c(\mathcal{G})} b_{\mathcal{G}, \mathbf{s}[\mathbf{V}^j]}^\top(\mathbf{v}^j, \mathsf{pa}_{\mathbf{V}^j}) \mathbf{p}_j, \mathbf{p}_j \in \Delta(|\Omega(\mathbf{M}^j)|) \right\}.$$

Space $\mathcal{M}_{\mathcal{G}}$ contains all interventional distribution tuples associated with canonical SCMs $\langle \mathbf{M}, \mathbf{V}, \mathbb{F}, P(\mathbf{M}) \rangle$ with arbitrary $P(\mathbf{M})$. Since the space of $\{\mathbf{p}_1, \ldots, \mathbf{p}_{n_c(\mathcal{G})}\}$ is compact and a continuous image of a compact space is compact, $\mathcal{M}_{\mathcal{G}}$ is also a compact space.

## 4 An Asymptotic Regret Lower Bound

In this section, we present an asymptotic information-theoretic regret lower bound for the causal bandit problem with confounders. It quantifies the optimal asymptotic performance of a uniformly good causal bandit policy defined below.

**Definition 3** (Uniformly Good Policy). *Given a causal graph $\mathcal{G} = \langle \mathbf{V}, \mathcal{E}, \mathcal{B} \rangle$, a causal bandit policy $\pi$ is uniformly good if for any $\alpha > 0$, the expected regret of $\pi$ for any interventional distribution tuple setup $\mathcal{P} \in \mathcal{M}_{\mathcal{G}}$, denoted by $R_T^\pi(\mathcal{P})$, satisfies*

$$\lim_{T \to \infty} R_T^\pi(\mathcal{P})/T^\alpha = 0.$$

A policy $\pi$ being uniformly good indicates that it can achieve subpolynomial regret for any $\mathcal{P} \in \mathcal{M}_{\mathcal{G}}$. The information-theoretic regret lower bound sets the limit for the asymptotic regret of all such policies. Let $D(P \parallel Q)$ denote the Kullback–Leibler (KL) divergence of two probability distributions $P$ and $Q$ and let $\mathcal{I}_{-\mathbf{s}} = \mathcal{I} \setminus \{\mathbf{s}\}$.

**Theorem 1.** *Given a causal graph $\mathcal{G} = \langle \mathbf{V}, \mathcal{E}, \mathcal{B} \rangle$, let $\mathcal{P} = \langle P_{\mathbf{s}} \rangle_{\mathbf{s} \in \mathcal{I}} \in \mathcal{M}_{\mathcal{G}}$ be the true interventional distribution tuple with a unique optimal intervention $\mathbf{s}_* \in \mathcal{I}$ with mean reward $\mu_*$. The expected regret for any uniformly good causal bandit policy $\pi$ satisfies*

$$\liminf_{T \to \infty} R_T^\pi(\mathcal{P})/\ln T \geq C(\mathcal{P}, \mathcal{G}),$$

*where $C(\mathcal{P}, \mathcal{G})$ is the value of the optimization problem given below,*

$$\mathbb{O}(\mathcal{P}, \mathcal{G}) : \min_{\eta_{\mathbf{s}} \geq 0, \forall \mathbf{s} \in \mathcal{I}_{-\mathbf{s}_*}} \sum_{\mathbf{s} \in \mathcal{I}_{-\mathbf{s}_*}} \eta_{\mathbf{s}} \Delta_{\mathbf{s}}, \tag{6}$$

$$\text{s.t.} \max_{\mathbf{s} \in \mathcal{I}_{-\mathbf{s}_*}} \mu_{\mathbf{s}}' \leq \mu_*, \forall \mathcal{P}' \in \left\{ \mathcal{P}'' \in \mathcal{M}_{\mathcal{G}} \,\Big|\, \sum_{\mathbf{s} \in \mathcal{I}_{-\mathbf{s}_*}} \eta_{\mathbf{s}} D(P_{\mathbf{s}} \parallel P_{\mathbf{s}}'') < 1, P_{\mathbf{s}_*}'' = P_{\mathbf{s}_*} \right\}, \tag{7}$$

*where $\mu_{\mathbf{s}}'$ is the expected reward of intervention $\mathbf{s}$ according to probability distribution $P_{\mathbf{s}}'$.*

Theorem 1 can be viewed as an extension of [10, Th. 1], and we defer its proof to appendices. The optimization problem $\mathbb{O}(P, \mathcal{M}_{\mathcal{G}})$ is a semi-infinite program since there are infinite $\mathcal{P}'$ in (7). To interpret the lower bound, (6) is a minimization of the regret, and its solution indicates an optimal allocation of $O(\ln T)$ number of explorations, which is $\eta_{\mathbf{s}} \ln T$ for each $\mathbf{s} \in \mathcal{I}_{-\mathbf{s}_*}$ as $T \to \infty$. In (7), $\sum_{\mathbf{s} \in \mathcal{I}} \eta_{\mathbf{s}} D(P_{\mathbf{s}} \parallel P_{\mathbf{s}}'')$ can be viewed as the distance generated between true interventional distribution tuple $\mathcal{P}$ and an alternative $\mathcal{P}'' \in \mathcal{M}_{\mathcal{G}}$ by an exploration allocation strategy $\langle \eta_{\mathbf{s}} \rangle_{\mathbf{s} \in \mathcal{I}_{-\mathbf{s}_*}}$. For any $\mathcal{P}'$ in (7), $\mu_{\mathbf{s}_*}' = \mu_*$ since $P_{\mathbf{s}_*}' = P_{\mathbf{s}_*}$, and as a result, $\max_{\mathbf{s} \in \mathcal{I}_{-\mathbf{s}_*}} \mu_{\mathbf{s}}' \leq \mu_*$ indicates intervention $\mathbf{s}_*$ is also optimal in $\mathcal{P}'$. So (7) sets a constraint for $\langle \eta_{\mathbf{s}} \rangle_{\mathbf{s} \in \mathcal{I}_{-\mathbf{s}_*}}$: the models allowed to have distance $< 1$ from the true model must also take $\mathbf{s}_*$ to be optimal. Or conversely, $\langle \eta_{\mathbf{s}} \rangle_{\mathbf{s} \in \mathcal{I}_{-\mathbf{s}_*}}$ must generate distance $\geq 1$ for any $\mathcal{P}' \in \mathcal{M}_{\mathcal{G}}$ with $P_{\mathbf{s}_*}' = P_{\mathbf{s}_*}$ not taking intervention $\mathbf{s}_*$ to be optimal.

**Remark 1.** *The $O(\ln T)$ regret lower bound holds only if there exists $\mathcal{P}' \in \mathcal{M}_{\mathcal{G}}$ such that $P_{\mathbf{s}_*}' = P_{\mathbf{s}_*}$, which is a condition meaning $\mathcal{P}$ can not be distinguished from all other interventional distribution tuples in $\mathcal{M}_{\mathcal{G}}$ by only taking intervention $\mathbf{s}_*$. This condition holds for almost all causal bandit problems. Otherwise, constraint (7) is relaxed, and the value of the optimization problem $C(\mathcal{P}, \mathcal{M}_{\mathcal{G}}) = 0$, which indicates sub-logarithmic regret can be achieved. Following the same line of proof in [17, Th. 3.9], it can be shown simple UCB can achieve this sub-logarithmic, in fact constant, expected regret.*

## 5 SCM-based Causal Bandit Algorithm

The regret lower bound in Theorem 1 suggests a potentially optimal exploration strategy, i.e. to take each intervention $\mathbf{s} \in \mathcal{I}_{-\mathbf{s}_*}$ up to $\eta_{\mathbf{s}} \ln T$ times. However, it requires the ground truth $\mathcal{P}$ to solve the optimization problem $\mathbb{O}(\mathcal{P}, \mathcal{G})$. The allocation matching principle [10] essentially replaces the true model with an estimated one to solve the optimization problem, and controls explorations to match with the solution. It is expected that the estimated model converges to the ground truth at a fast enough rate so that the actual explorations match with the optimal exploration strategy. We follow this idea to design a causal bandit policy, whose pseudo-code is shown in Algorithm 1. We call this SCM-based Approximate Allocation Matching (SCM-AAM).

Similar to [22], SCM-AAM only intervenes on POMISs for the purpose of reducing the action space. The algorithm is anytime without requiring the knowledge of horizon length. It takes two tuning parameters $\epsilon = (0, 1/|\mathcal{I}|)$ and $\gamma > 0$ as inputs that control the exploration rate, which affects finite-time performance. At each time $t$, let $N_t(\mathbf{s})$ be the number of times the intervention $do(\mathbf{S} = \mathbf{s})$ is taken so far, and let $N_t(\mathbf{v}, \mathbf{s})$ be the number of times we observe $\mathbf{v}$ with intervention $do(\mathbf{S} = \mathbf{s})$. The algorithm maintains a set of empirical interventional distributions $\langle \bar{P}_{\mathbf{s},t} \rangle_{\mathbf{s} \in \mathcal{I}}$ and empirical mean rewards $\langle \bar{\mu}_{\mathbf{s},t} \rangle_{\mathbf{s} \in \mathcal{I}}$, where $\bar{P}_{\mathbf{s},t}(\mathbf{v}) = N_t(\mathbf{v}, \mathbf{s})/N_t(\mathbf{s})$ for each $\mathbf{s} \in \mathcal{I}$.

The main body of the algorithm is composed of two components: exploitation and exploration. At each round $t$, it first attempts to evaluate if enough information has been collected to determine the optimal action. Inspired by the lower bound, enough distance is generated by the sampling history between $\langle \bar{P}_{\mathbf{s},t} \rangle_{\mathbf{s} \in \mathcal{I}}$ and $\mathcal{P}' = \langle P_{\mathbf{s}}' \rangle_{\mathbf{s} \in \mathcal{I}}$ if $\sum_{\mathbf{s} \in \mathcal{I}} N_t(\mathbf{s}) D(\bar{P}_{\mathbf{s},t} \parallel P_{\mathbf{s}}') > (1 + \gamma) \ln t$. So we can construct a confidence set $\mathcal{C}_t$ with high probability to contain the true model $\mathcal{P}$, which is defined as

$$\mathcal{C}_t := \left\{ \mathcal{P}' \in \mathcal{M}_{\mathcal{G}} \,\Big|\, \sum_{\mathbf{s} \in \mathcal{I}} N_t(\mathbf{s}) D(\bar{P}_{\mathbf{s},t} \parallel P_{\mathbf{s}}') \leq (1 + \gamma) \ln t \right\}. \tag{8}$$

**Algorithm 1:** SCM-based Approximate Allocation Matching (SCM-AAM)

**Input**         : Causal graph $\mathcal{G} = \langle \mathbf{V}, \mathcal{E}, \mathcal{B} \rangle$, $\mathcal{I} = \{\mathbf{s} \in \Omega(\mathbf{S}) \mid \mathbf{S} \text{ is POMIS}\}$
**Initialization**: $N_t^{\mathrm{e}} = 0$, $\forall \mathbf{s} \in \mathcal{I} : N_t(\mathbf{s}) = 0$
**Set**           : $\epsilon \in (0, 1/|\mathcal{I}|)$ and $\gamma > 0$
**Output**        : Sequence of interventions

Select each intervention $\mathbf{s} \in \mathcal{I}$ once
**for** $t = |\mathcal{I}| + 1$ **to** $T$ **do**
    Compute empirical distributions and mean rewards $\langle \bar{P}_{\mathbf{s},t} \rangle_{\mathbf{s} \in \mathcal{I}}$ and $\langle \bar{\mu}_{\mathbf{s},t} \rangle_{\mathbf{s} \in \mathcal{I}}$
    With $\mathcal{C}_t$ defined in (8), compute UCB $\tilde{\mu}_{\mathbf{s}}(\mathcal{C}_t) = \sum_{\mathbf{v} \in \Omega(\mathbf{V})} v_n \max_{\mathcal{P}' \in \mathcal{C}_t} P'_{\mathbf{s}}(\mathbf{v})$, $\forall \mathbf{s} \in \mathcal{I}$

1     **if** $\exists \mathbf{s}_{*,t} : \bar{\mu}_{\mathbf{s}_{*,t}} > \tilde{\mu}_{\mathbf{s}}(\mathcal{C}_t), \forall \mathbf{s} \in \mathcal{I}_{-\mathbf{s}_{*,t}}$ **then**
        Select $\mathbf{S}_t \leftarrow \mathbf{s}_{*,t}$, and $N_t(\mathbf{S}_t) \leftarrow N_t(\mathbf{S}_t) + 1$ *% exploitation*

2     **else if** $\min_{\mathbf{s} \in \mathcal{I}} N_t(\mathbf{s}) \leq \epsilon N_t^{\mathrm{e}}$ **then**
        Select $\underline{\mathbf{S}}_t \leftarrow \arg\min_{\mathbf{s} \in \mathcal{I}} N_t(\mathbf{s})$ *% forced exploration*
        Set $N_t^{\mathrm{e}} \leftarrow N_t^{\mathrm{e}} + 1$ and $N_t(\underline{\mathbf{S}}_t) \leftarrow N_t(\underline{\mathbf{S}}_t) + 1$

3     **else**
        Select $\overline{\mathbf{S}}_t \leftarrow \arg\max_{\mathbf{s} \in \mathcal{I}} \tilde{\mu}_{\mathbf{s}}(\mathcal{C}_t)$ *% exploration with greedy approximation*
        Set $N_t^{\mathrm{e}} \leftarrow N_t^{\mathrm{e}} + 1$ and $N_t(\overline{\mathbf{S}}_t) \leftarrow N_t(\overline{\mathbf{S}}_t) + 1$

---

With $V_n$ being the reward node, for each $\mathbf{s} \in \mathcal{I}$, the algorithm computes the UCB of the mean reward of taking intervention $\mathbf{s}$ as $\tilde{\mu}_{\mathbf{s}}(\mathcal{C}_t) = \sum_{\mathbf{v} \in \Omega(\mathbf{V})} v_n \max_{\mathcal{P}' \in \mathcal{C}_t} P'_{\mathbf{s}}(\mathbf{v})$. If there exists an intervention $\mathbf{s}_{*,t}$ with empirical mean reward $\bar{\mu}_{\mathbf{s}_{*,t}} > \tilde{\mu}_{\mathbf{s}}(\mathcal{C}_t), \forall \mathbf{s} \in \mathcal{I}_{-\mathbf{s}_{*,t}}$, we are confident about $\mathbf{s}_{*,t}$ to be the optimal intervention. So the algorithm enters the exploitation phase and selects $\mathbf{S}_t = \mathbf{s}_{*,t}$.

If no such $\mathbf{s}_{*,t}$ exists, the algorithm needs to explore. According to the allocation principle, one is inclined to solve $\mathbb{O}(\langle \bar{P}_{\mathbf{s},t} \rangle_{\mathbf{s} \in \mathcal{I}}, \mathcal{G})$ defined in Theorem 1 and make $N_t(\mathbf{s}) \approx \eta_{\mathbf{s}} \ln t$ for each $\mathbf{s} \in \mathcal{I}$. Nevertheless, there is no guarantee that this semi-infinite program can be solved efficiently. Therefore, we take a greedy approximation approach. Ideally, to get out of exploration, the algorithm needs to reduce $\max_{\mathbf{s} \in \mathcal{I}_{-\mathbf{s}_*}} \tilde{\mu}_{\mathbf{s}}(\mathcal{C}_t)$ to a value below $\mu_*$, thus it selects $\overline{\mathbf{S}}_t = \arg\max_{\mathbf{s} \in \mathcal{I}} \tilde{\mu}_{\mathbf{s}}(\mathcal{C}_t)$. The algorithm maintains a counter of exploration steps $N_t^{\mathrm{e}}$. If $\min_{\mathbf{s} \in \mathcal{I}} N_t(\mathbf{s}) \leq \epsilon N_t^{\mathrm{e}}$, it generates forced exploration by selecting the least selected intervention $\underline{\mathbf{S}}_t$. Such a mechanism ensures every action is persistently taken in exploration phases so that $\langle \bar{P}_{\mathbf{s},t} \rangle_{\mathbf{s} \in \mathcal{I}}$ converges to the ground truth eventually.

**Remark 2.** *With factorization in* (5), $\mathcal{C}_t$ *corresponds to setting constraint for* $\{\mathbf{p}_1, \ldots, \mathbf{p}_{n_{\mathrm{c}}(\mathcal{G})}\}$ *as*

$$\sum_{\mathbf{s} \in \mathcal{I}} \sum_{\mathbf{v} \in \Omega(\mathbf{V})} N_t(\mathbf{s}, \mathbf{v}) \ln \frac{\bar{P}_{\mathbf{s},t}(\mathbf{v})}{\prod_{j=1}^{n_{\mathrm{c}}(\mathcal{G})} b_{\mathcal{G}, \mathbf{s}[\mathbf{V}^j]}^{\top}(\mathbf{v}^j, \mathsf{pa}_{\mathbf{V}^j}) \mathbf{p}_j} \leq (1 + \gamma) \ln t, \tag{9}$$

*where the expression on the left of inequality is a convex function of* $\mathbf{p}_1, \ldots, \mathbf{p}_{n_{\mathrm{c}}(\mathcal{G})}$. *Define convex set* $\mathcal{C}_t^{\mathrm{p}} = \{\{\mathbf{p}_1, \ldots, \mathbf{p}_{n_{\mathrm{c}}(\mathcal{G})}\} \mid \mathbf{p}_j \in \Delta(|\Omega(\mathbf{M}^j)|), \forall j \in \{1, \ldots, n_{\mathrm{c}}(\mathcal{G})\}, \text{ and } (9) \text{ is true}\}$. *Then the UCB can be derived by maximizing a set of concave functions over* $\mathcal{C}_t^{\mathrm{p}}$ *as follows,*

$$\tilde{\mu}_{\mathbf{s}}(\mathcal{C}_t) = \sum_{\mathbf{v} \in \Omega(\mathbf{V})} v_n \max_{\mathcal{P}' \in \mathcal{C}_t} P'_{\mathbf{s}}(\mathbf{v}) = \sum_{\mathbf{v} \in \Omega(\mathbf{V})} v_n \exp\left( \max_{\{\mathbf{p}_1, \ldots, \mathbf{p}_{n_{\mathrm{c}}(\mathcal{G})}\} \in \mathcal{C}_t^{\mathrm{p}}} \sum_{j=1}^{n_{\mathrm{c}}(\mathcal{G})} \ln\left( b_{\mathcal{G}, \mathbf{s}[\mathbf{V}^j]}^{\top}(\mathbf{v}^j, \mathsf{pa}_{\mathbf{V}^j}) \mathbf{p}_j \right) \right).$$

*We use* $\tilde{\mu}_{\mathbf{s}}(\mathcal{C}_t)$ *instead of* $\tilde{\mu}'_{\mathbf{s}}(\mathcal{C}_t) = \max_{\mathcal{P}' \in \mathcal{C}_t} \sum_{\mathbf{v} \in \Omega(\mathbf{V})} v_n P'_{\mathbf{s}}(\mathbf{v})$ *as UCB since* $\sum_{\mathbf{v} \in \Omega(\mathbf{V})} v_n P'_{\mathbf{s}}(\mathbf{v})$ *in general is a non-concave function of* $\mathbf{p}_1, \ldots, \mathbf{p}_{n_{\mathrm{c}}(\mathcal{G})}$. *Also, notice that* $\forall \mathbf{s} \in \mathcal{I} : \tilde{\mu}_{\mathbf{s}}(\mathcal{C}_t) \geq \tilde{\mu}'_{\mathbf{s}}(\mathcal{C}_t)$.

In a pure topology-based approach [22], each intervention on POMISs is treated independently during the algorithm execution. Whereas SCM-AAM can leverage the canonical SCM to incorporate the sampling results from different interventions together. After each new sample $\mathbf{v}_t$ with intervention $\mathbf{s}_t$ is observed, it is used to update the parametric space for interventional distribution tuples $\mathcal{C}_t$.

## 6  Finite Time Regret Analysis

We present a finite time problem-dependent regret upper bound for SCM-AAM in Theorem 2. The complete proof is provided in the appendices. We also give an interpretation of Theorem 2, and compare the regret upper bound with [22] in the setting where the reward node $V_n$ is binary,

**Theorem 2.** *For the causal bandit problem with unobserved confounders, suppose the causal graph $\mathcal{G} = \langle \mathbf{V}, \mathcal{E}, \mathcal{B} \rangle$ is given and Assumptions 1 and 2 are true. For any interventional distribution tuple $\mathcal{P} \in \mathcal{M}_{\mathcal{G}}$, any $\kappa > 0$ and horizon of length $T$, the expected regret for SCM-AAM with parameters $\epsilon \in (0, 1/|\mathcal{I}|)$ and $\gamma > 0$ satisfies*

$$R_T^{\text{SCM-AAM}}(\mathcal{P}) \leq (1+\kappa)(1+\gamma) \sum_{\mathbf{s} \in \mathcal{I}_{-\mathbf{s}_*}} \zeta_{\mathbf{s}}(\mathcal{P})\Big(\Delta_{\mathbf{s}} + \frac{\epsilon|\mathcal{I}|}{1 - \epsilon|\mathcal{I}|}\Big)\ln T + c,$$

*where $c$ is a suitably large universal constant depending on $\mathcal{P}$, $\kappa$ and tuning parameters $\epsilon$ and $\gamma$, and*

$$\forall \mathbf{s} \in \mathcal{I}_{-\mathbf{s}_*} : \zeta_{\mathbf{s}}(\mathcal{P}) = \begin{cases} 0, & \tilde{\mu}_{\mathbf{s}}(\mathcal{C}_{\mathbf{s}}(0, \mathcal{P})) \leq \mu_*, \\ \max_{\eta_{\mathbf{s}} \geq 0} \eta_{\mathbf{s}} \;\; \text{s.\,t.} \;\; \tilde{\mu}_{\mathbf{s}}(\mathcal{C}_{\mathbf{s}}(\eta_{\mathbf{s}}, \mathcal{P})) \geq \mu_*, & \text{otherwise}, \end{cases}$$

*in which $\mathcal{C}_{\mathbf{s}}(\eta_{\mathbf{s}}, \mathcal{P}) := \big\{ \mathcal{P}' \in \mathcal{M}_{\mathcal{G}} \mid \eta_{\mathbf{s}} D(P_{\mathbf{s}} \parallel P'_{\mathbf{s}}) + \epsilon \eta_{\mathbf{s}}/2 \sum_{\mathbf{x} \in \mathcal{I}_{-\mathbf{s}}} D(P_{\mathbf{x}} \parallel P'_{\mathbf{x}}) \leq 1 \big\}$.*

Theorem 2 indicates the expected number of samples for each $\mathbf{s} \in \mathcal{I}_{-\mathbf{s}_*}$ can be approximately bounded by $\zeta_{\mathbf{s}}(\mathcal{P}) \ln T$. It can be interpreted as follows: when $\mathbf{s} \in \mathcal{I}_{-\mathbf{s}_*}$ is selected approximately up to $\zeta_{\mathbf{s}}(\mathcal{P}) \ln T$ times, its UCB can be pushed below $\mu_*$. The presence of terms containing $\epsilon$ is due to forced exploration. In [22], KL-UCB and Thompson sampling are employed to apply interventions on POMISs. If the reward node $V_n \in \{0, 1\}$, they enjoys expected regret $\sum_{\mathbf{s} \in \mathcal{I}_{-\mathbf{s}_*}} (\Delta_{\mathbf{s}} \ln T)/D(P_{\mathbf{s}}(V_n) \parallel P_{\mathbf{s}_*}(V_n)) + c'$ for some insignificant $c'$ [14, 2]. In this situation, Theorem 2 shows that the SCM-AAM enjoys smaller expected regret when we disregard the small constants $\kappa$, $\gamma$ and $\epsilon$. To see it, we express the leading constant for KL-UCB and Thompson as

$$1/D(P_{\mathbf{s}}(V_n) \parallel P_{\mathbf{s}_*}(V_n)) = \max_{\eta_{\mathbf{s}} \geq 0} \eta_{\mathbf{s}} \;\; \text{s.\,t.} \;\; \tilde{\mu}_{\mathbf{s}}(\mathcal{C}'_{\mathbf{s}}(\eta_{\mathbf{s}}, P(V_n))) \geq \mu_*,$$

where $\mathcal{C}'_{\mathbf{s}}(\eta_{\mathbf{s}}, P(V_n)) = \{P'_{\mathbf{s}}(V_n) \in [0, 1] \mid \eta_{\mathbf{s}} D(P_{\mathbf{s}}(V_n) \parallel P'_{\mathbf{s}}(V_n)) \leq 1\}$. Note that $\mathcal{C}'_{\mathbf{s}}(\eta_{\mathbf{s}}, P(V_n))$ only sets constraint on $P'_{\mathbf{s}}(V_n)$, while $\mathcal{C}_{\mathbf{s}}(\eta_{\mathbf{s}}, \mathcal{P})$ sets constraints on $P'_{\mathbf{s}}(\mathbf{V})$. As a result, $\mathcal{C}_{\mathbf{s}}(\eta_{\mathbf{s}}, \mathcal{P}) \subseteq \mathcal{C}'_{\mathbf{s}}(\eta_{\mathbf{s}}, P(V_n))$, so that $\zeta_{\mathbf{s}}(\mathcal{P}) \leq 1/D(P_{\mathbf{s}}(V_n) \parallel P_{\mathbf{s}_*}(V_n))$.

## 7  Experimental Results

We compare the empirical performance of SCM-AAM with existing algorithms. The first baseline algorithm employs the simple UCB algorithm [4] on a reduced action set that includes interventions on POMISs. The other two baselines as introduced in [22], utilize the KL-UCB algorithm and Thomposn sampling (TS) to intervene on POMISs. We compare the performance of all the algorithms on three different causal bandit instances shown in Fig. 2. We choose the input parameters of the SCM-AAM algorithm to be $\gamma = 0.1$ and $\epsilon = 1/|5\mathcal{I}|$ in the simulations. We set the horizon to $800$ for all three tasks and repeat every simulation $100$ times. Additionally, we include a confidence interval around the mean regret, with the width of the interval set to twice the standard deviation. The structural equations for the three causal bandit instances, as well as experimental details, can be found in the appendices. We plot the mean regret against time, where time in this context corresponds to the number of actions. All the nodes in Fig. 2 take binary values, either 0 or 1.

For task 1, we consider the causal graph shown in Fig. 2(a) with POMISs $\{V_1\}$ and $\{V_2\}$. For task 2 and task 3, we consider more complex causal graphs, as displayed in Fig. 2(b) and (c). The POMISs for task 2 include $\{V_1\}$, $\{V_3\}$, and $\{V_3, V_4\}$, while for task 3, they are $\{V_1, V_4, V_6\}$ and $\{V_5, V_6\}$. The UCB algorithm has inferior performance compared to other baseline algorithms, especially in the more complicated tasks 2 and 3. The TS algorithm outperforms both UCB and KL-UCB in all three settings; however, it still incurs more regret than our proposed algorithm. The experiments demonstrate that our proposed SCM-AAM algorithm consistently outperforms other baseline algorithms by incurring lower empirical regret across all three tasks. Notably, the performance advantage of SCM-AAM is more significant for tasks 2 and 3, which have a more intricate causal graph structure with a higher number of nodes and edges. The results indicate that

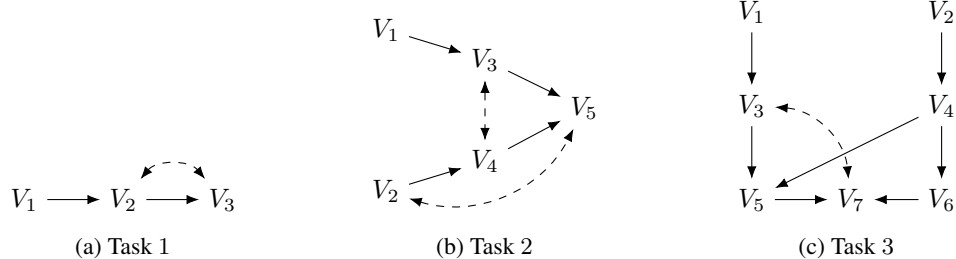

(a) Task 1            (b) Task 2            (c) Task 3

Figure 2: Causal graphs used in the experiments

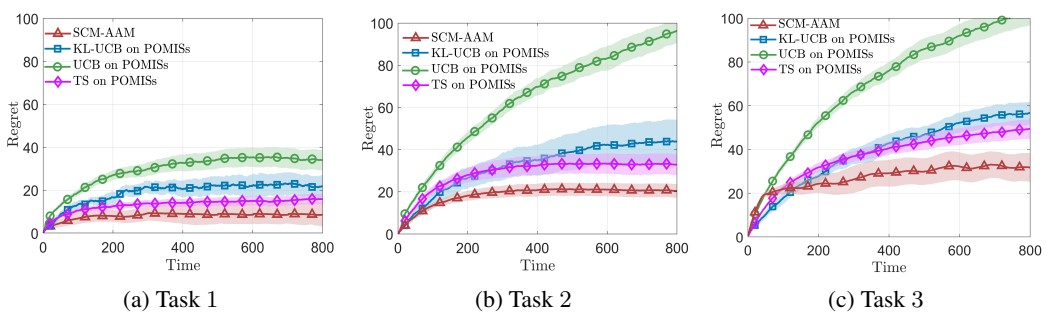

(a) Task 1            (b) Task 2            (c) Task 3

Figure 3: Regret versus time for the instances of structural causal bandits shown in Figure 2

incorporating causal structural information through the SCM-AAM algorithm can significantly reduce expected regret, especially for more complex causal graphs.

The simulations were conducted on a Windows desktop computer featuring a 12th generation Intel Core i7 processor operating at 3.1 GHz and 32 GB of RAM. No GPUs were utilized in the simulations. The runtime for each iteration of the SCM-MAB, involving 800 arm plays, was approximately 2 minutes for both Task 1. However, for Task 2 and Task 3, which encompass larger underlying causal graphs, the runtime for every iteration increased to around 20 minutes. The algorithm code is provided at https://github.com/CausalML-Lab/SCM-AAM.

## 8 Conclusion

In this paper, we studied the causal bandit problem with latent confounders. With causal information provided by a causal graph, we present a problem-dependent information-theoretic regret lower bound. Leveraging the canonical SCM, we take an approximate allocation matching strategy to design the SCM-AAM algorithm. By analyzing SCM-AAM, we show it has a problem-dependent logarithmic regret upper bound. The analytic result is complemented with numerical illustrations featuring a variety of causal structures. It is shown that SCM-AAM exhibits promising performance in comparison with classic baseline algorithms. Since SCM has a natural application in counterfactual reasoning, extending the proposed algorithm and theoretical results to a counterfactual decision-making setup is an interesting future direction.

## Acknowledgements

This research has been supported in part by NSF CAREER 2239375. Most of the work was completed when Lai Wei was a postdoctoral researcher at Purdue University.

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

# Appendices

## A  Proof of Asymptotic Regret Lower Bound in Theorem 1

The lower bound is derived following the same strategy in [21] by applying divergence decomposition and Bretagnolle–Huber inequality. For completeness, we reproduce both proofs in this section. Readers familiar with these results can skip them. Besides, we note that a more general result on controlled Markov chains is proven in [15].

Recall that a policy $\pi$ is composed of a sequence $\{\pi_t\}_{t \in \mathbb{N}_{>0}}$, where at each time $t \in \{1, \ldots, T\}$, $\pi_t$ determines the probability distribution of taking intervention $\mathbf{S}_t \in \mathcal{I}$ given intervention and observation history $\pi_t(\mathbf{S}_t \mid \mathbf{s}_1, \mathbf{v}_1, \ldots, \mathbf{s}_{t-1}, \mathbf{v}_{t-1})$. So the intervention and observation sequence $\{\mathbf{S}_t, \mathbf{V}_t\}_{t \in \mathbb{N}_{>0}}$ is a production of the interactions between the interventional distribution tuple $\langle P_\mathbf{s} \rangle_{\mathbf{s} \in \mathcal{I}}$ and policy $\pi$. Let $T$ be the horizon length, we define a probability measure $\mathbb{P}$ on the sequence of outcomes induced by $\langle P_\mathbf{s} \rangle_{\mathbf{s} \in \mathcal{I}}$ and $\pi$ such that

$$\mathbb{P}(\mathbf{s}_1, \mathbf{v}_1, \ldots, \mathbf{s}_T, \mathbf{v}_T) = \prod_{t=1}^{T} \pi_t(\mathbf{s}_t \mid \mathbf{s}_1, \mathbf{v}_1, \ldots, \mathbf{s}_{t-1}, \mathbf{v}_{t-1}) P_{\mathbf{s}_t}(\mathbf{v}_t). \tag{10}$$

For a fixed policy $\pi$, let $\mathbb{P}$ and $\mathbb{P}'$ be the probability measures on the $T$-round plays of two different causal bandit instances with different interventional distribution tuples $\mathcal{P} = \langle P_\mathbf{s} \rangle_{\mathbf{s} \in \mathcal{I}}$ and $\mathcal{P}' = \langle P_\mathbf{s}' \rangle_{\mathbf{s} \in \mathcal{I}}$ in space $\mathcal{M}_\mathcal{G}$, which is defined in Definition 2. From (10), we get the distinction between $\mathbb{P}$ and $\mathbb{P}'$ is exclusively due to the separations of $\mathcal{P}$ and $\mathcal{P}'$. As a result, $D(\mathbb{P} \parallel \mathbb{P}')$ has the following decomposition, which is a standard result in MAB problems [21, Th. 15.1].

**Lemma 3** (Divergence Decomposition). *Given a causal graph $\mathcal{G} = \langle \mathbf{V}, \mathcal{E}, \mathcal{B} \rangle$, let $\mathcal{P} = \langle P_\mathbf{s} \rangle_{\mathbf{s} \in \mathcal{I}}$ be the true interventional distribution tuple and let $\mathcal{P}' = \langle P_\mathbf{s}' \rangle_{\mathbf{s} \in \mathcal{I}}$ be different from $\mathcal{P}$. For some fixed policy $\pi$, let $\mathbb{P}$ and $\mathbb{P}'$ be the probability measures on the $T$ rounds of causal bandit plays in $\mathcal{P}$ and $\mathcal{P}'$ respectively. Then,*

$$D(\mathbb{P} \parallel \mathbb{P}') = \sum_{\mathbf{s} \in \mathcal{I}} \mathbb{E}[N_T(\mathbf{s})] D\left(P_\mathbf{s} \parallel P_\mathbf{s}'\right),$$

*where the expectation is computed with probability measure $\mathbb{P}$.*

*Proof.* For a fixed policy $\pi$, from (10), we get

$$\mathbb{P}(\mathbf{s}_1, \mathbf{v}_1, \ldots, \mathbf{s}_T, \mathbf{v}_T) = \prod_{t=1}^{T} \pi_t(\mathbf{s}_t \mid \mathbf{s}_1, \mathbf{v}_1, \ldots, \mathbf{s}_{t-1}, \mathbf{v}_{t-1}) P_{\mathbf{s}_t}(\mathbf{v}_t).$$

As a result, $\pi_t$ is reduced and we get

$$\log \frac{\mathbb{P}(\mathbf{s}_1, \mathbf{v}_1, \ldots, \mathbf{s}_T, \mathbf{v}_T)}{\mathbb{P}'(\mathbf{s}_1, \mathbf{v}_t, \ldots, \mathbf{s}_T, \mathbf{v}_T)} = \sum_{t=1}^{T} \log \frac{P_{\mathbf{s}_t}(\mathbf{v}_t)}{P'_{\mathbf{s}_t}(\mathbf{v}_t)}, \tag{11}$$

which shows the distinction between $\mathbb{P}$ and $\mathbb{P}'$ is exclusively due to the separations of $P_\mathbf{s}$ and $P_\mathbf{s}'$ for each $\mathbf{s} \in \mathcal{I}$. Then we can decompose $D(\mathbb{P} \parallel \mathbb{P}')$ with (11) as the following,

$$D(\mathbb{P} \parallel \mathbb{P}') = \mathbb{E}\left[\log \frac{\mathbb{P}(\mathbf{S}_1, \mathbf{V}_1, \ldots, \mathbf{S}_T, \mathbf{V}_T)}{\mathbb{P}'(\mathbf{S}_1, \mathbf{V}_t, \ldots, \mathbf{S}_T, \mathbf{V}_T)}\right] = \mathbb{E}\left[\sum_{t=1}^{T} \log \frac{P_{\mathbf{S}_t}(\mathbf{V}_t)}{P'_{\mathbf{S}_t}(\mathbf{V}_t)}\right] = \sum_{t=1}^{T} \mathbb{E}\left[\log \frac{P_{\mathbf{S}_t}(\mathbf{V}_t)}{P'_{\mathbf{S}_t}(\mathbf{V}_t)}\right]$$

$$= \sum_{t=1}^{T} \mathbb{E}\left[\mathbb{E}\left[\log \frac{P_{\mathbf{S}_t}(\mathbf{V}_t)}{P'_{\mathbf{S}_t}(\mathbf{V}_t)} \Big| \mathbf{S}_t\right]\right] = \sum_{t=1}^{T} \mathbb{E}\left[D\left(P_{\mathbf{S}_t} \parallel P'_{\mathbf{S}_t}\right)\right]$$

$$= \sum_{t=1}^{T} \mathbb{E}\left[\sum_{\mathbf{s} \in \mathcal{I}} \mathbb{1}\{\mathbf{S}_t = \mathbf{s}\} D\left(P_\mathbf{s} \parallel P_\mathbf{s}'\right)\right] = \sum_{\mathbf{s} \in \mathcal{I}} D\left(P_\mathbf{s} \parallel P_\mathbf{s}'\right) \mathbb{E}\left[\sum_{t=1}^{T} \mathbb{1}\{\mathbf{S}_t = \mathbf{s}\}\right]$$

$$= \sum_{a \in \mathcal{I}} \mathbb{E}[N_T(\mathbf{s})] D\left(P_\mathbf{s} \parallel P_\mathbf{s}'\right),$$

which concludes the proof. $\qquad \square$

The other tool to prove the regret lower bound is the Bretagnolle–Huber Inequality.

**Lemma 4** (Bretagnolle–Huber Inequality [21, Th. 14.2])**.** *Let $P$ and $Q$ be two probability measures on a measurable space $(\Omega, \mathcal{F})$, and let $E \in \mathcal{F}$ be an arbitrary event. Then*

$$P(E) + Q(E^{\complement}) \geq \frac{1}{2} \exp\left(-D(P \parallel Q)\right),$$

*where $E^{\complement} = \Omega \setminus E$ is complement of $E$.*

*Proof with finite $\Omega$.* In Assumption 1, we assume each node in the causal graph can only take a finite number of values. So the sample space for the causal bandit problem with horizon length $T$ is finite. We provide proof for the Bretagnolle–Huber inequality assuming $\Omega$ is finite. First, we show

$$P(E) + Q(E^{\complement}) = \sum_{\omega \in E} P(\omega) + \sum_{\omega \in E^{\complement}} Q(\omega) \geq \sum_{\omega \in E} \min\{P(\omega), Q(\omega)\} + \sum_{\omega \in E^{\complement}} \min\{P(\omega), Q(\omega)\}$$

$$= \sum_{\omega \in \Omega} \min\{P(\omega), Q(\omega)\} = \frac{1}{2} \sum_{\omega \in \Omega} \min\{P(\omega), Q(\omega)\} \sum_{\omega \in \Omega} [P(\omega) + Q(\omega)]$$

$$\geq \frac{1}{2} \sum_{\omega \in \Omega} \min\{P(\omega), Q(\omega)\} \sum_{\omega \in \Omega} \max\{P(\omega), Q(\omega)\}.$$

Applying Cauchy–Schwarz inequality, we get

$$P(E) + Q(E^{\complement})$$

$$\geq \frac{1}{2} \left( \sum_{\omega \in \Omega} \sqrt{\min\{P(\omega), Q(\omega)\} \max\{P(\omega), Q(\omega)\}} \right)^2 = \frac{1}{2} \left( \sum_{\omega \in \Omega} \sqrt{P(\omega)Q(\omega)} \right)^2$$

$$= \frac{1}{2} \exp\left( 2 \log \sum_{\omega \in \Omega} \sqrt{P(\omega)Q(\omega)} \right) = \frac{1}{2} \exp\left( 2 \log \sum_{\omega \in \Omega} P(\omega) \sqrt{\frac{Q(\omega)}{P(\omega)}} \right)$$

$$\geq \frac{1}{2} \exp\left( 2 \sum_{\omega \in \Omega} P(\omega) \log \sqrt{\frac{Q(\omega)}{P(\omega)}} \right) = \frac{1}{2} \exp\left(-D(P \parallel Q)\right).$$

We conclude our proof. $\qquad\square$

To prove the regret lower bound, the remaining work regards applying divergence decomposition and substituting $D(\mathbb{P} \parallel \mathbb{P}')$ into the Bretagnolle–Huber inequality.

**Lemma 5.** *Given a causal graph $\mathcal{G} = \langle \mathbf{V}, \mathcal{E}, \mathcal{B} \rangle$, let $\mathcal{P} = \langle P_{\mathbf{s}} \rangle_{\mathbf{s} \in \mathcal{I}} \in \mathcal{M}_{\mathcal{G}}$ be the true interventional distribution tuple with a unique optimal intervention $\mathbf{s}_* \in \mathcal{I}$ with mean reward $\mu_*$. The expected regret for any uniformly good causal bandit policy $\pi$ satisfies*

$$\liminf_{T \to \infty} \frac{R_T^{\pi}(\mathcal{P})}{\ln T} \geq C(\mathcal{P}, \mathcal{G}),$$

*where $C(\mathcal{P}, \mathcal{M}_{\mathcal{G}})$ is the value of the optimization problem given below*

$$\tilde{\mathbb{O}}(\mathcal{P}, \mathcal{M}_{\mathcal{G}}): \underset{\eta_{\mathbf{s}} \geq 0, \forall \mathbf{s} \in \mathcal{I}_{-\mathbf{s}_*}}{\text{minimize}} \sum_{\mathbf{s} \in \mathcal{I}_{-\mathbf{s}_*}} \eta_{\mathbf{s}} \Delta_{\mathbf{s}} \tag{12}$$

$$\text{s.t.} \sum_{\mathbf{s} \in \mathcal{I}} \eta_{\mathbf{s}} D(P_{\mathbf{s}} \parallel \mathcal{P}'_{\mathbf{s}}) \geq 1, \forall P' \in \left\{ \mathcal{P}'' \in \mathcal{M}_{\mathcal{G}} \mid \mathbf{s}_* \notin \mathcal{I}''_* \right\}, \tag{13}$$

*where $\mathcal{I}''_*$ is the set of optimal interventions according to $\mathcal{P}''$.*

**Remark 3.** *To interpret Lemma 5, (12) is a minimize of the regret, and its solution indicates an optimal exploration strategy. Each $\mathbf{s} \in \mathcal{I}_{-\mathbf{s}_*}$ should be selected up to $\eta_{\mathbf{s}} \ln T$ as $T \to \infty$. In (13), $\sum_{\mathbf{s} \in \mathcal{I}} \eta_{\mathbf{s}} D(P_{\mathbf{s}} \parallel P''_{\mathbf{s}})$ is viewed as the distance generated between true interventional distribution tuple $\mathcal{P}$ and an alternative $\mathcal{P}'' \in \mathcal{M}_{\mathcal{G}}$ by an exploration strategy $\langle \eta_{\mathbf{s}} \rangle_{\mathbf{s} \in \mathcal{I}_{-\mathbf{s}_*}}$. So (13) requires $\langle \eta_{\mathbf{s}} \rangle_{\mathbf{s} \in \mathcal{I}_{-\mathbf{s}_*}}$ must generate distance $\geq 1$ for any $\mathcal{P}' \in \mathcal{M}_{\mathcal{G}}$ in which intervention $\mathbf{s}_*$ is not optimal.*

*Proof.* For any $\mathcal{P}' \in \{\mathcal{P}'' \in \mathcal{M}_\mathcal{G} \mid \mathbf{s}_* \notin \mathcal{I}''_*\}$, the optimal intervention $\mathbf{s}_*$ according to ground truth $\mathcal{P}$ is not optimal in $\mathcal{P}'$. Recall $\mathbb{P}'$ is the measures on $T$-round plays of causal bandit with interventional distribution tuple $\mathcal{P}'$. Also let $\mu'_*$ be the optimal mean reward and $\Delta'_\mathbf{s} = \mu'_* - \mu'_\mathbf{s}$. For true causal model $\mathcal{P}$, $\mathbb{P}$ and $\Delta_\mathbf{s}$ are defined similarly. For a fixed policy $\pi$, let $R_T$ and $R'_T$ be the expected regret from $T$ round plays in $\mathcal{P}$ and $\mathcal{P}'$ respectively. Letting $\epsilon = \min\{\Delta_{\mathbf{s}'_*}, \Delta'_{\mathbf{s}_*}\}$, we get

$$R_T \geq \frac{\epsilon T}{2} \mathbb{P}\Big(N_T(\mathbf{s}_*) < \frac{T}{2}\Big) \text{ and } R'_T \geq \frac{\epsilon T}{2} \mathbb{P}'\Big(N_T(\mathbf{s}_*) \geq \frac{T}{2}\Big).$$

Combining both inequalities, we get

$$\mathbb{P}\Big(N_T(\mathbf{s}_*) < \frac{T}{2}\Big) + \mathbb{P}'\Big(N_T(\mathbf{s}_*) \geq \frac{T}{2}\Big) \leq \frac{2}{\epsilon T}(R_T + R'_T).$$

Applying Bretagnolle–Huber inequality in Lemma 4 by substituting event $\{N_T(\mathbf{s}_*) < \frac{T}{2}\}$ into $E$,

$$D(\mathbb{P} \parallel \mathbb{P}') \geq \ln \frac{\epsilon T}{4(R_T + R'_T)}. \tag{14}$$

With Lemma 3, we substitute $\sum_{\mathbf{s} \in \mathcal{I}} \mathbb{E}[N_T(\mathbf{s})] D(P_\mathbf{s} \parallel P'_\mathbf{s})$ into $D(\mathbb{P} \parallel \mathbb{P}')$ and rearrange (14),

$$\frac{\sum_{\mathbf{s} \in \mathcal{I}} \mathbb{E}[N_T(\mathbf{s})] D(P_\mathbf{s} \parallel P'_\mathbf{s})}{\ln T} \geq 1 - \frac{\ln 4(R_T + R'_T)/\epsilon}{\ln T}. \tag{15}$$

According to Definition 3, that for any $\alpha > 0$, the regret for a uniformly good policy $\pi$ satisfies

$$\lim_{T \to \infty} \frac{R_T}{T^\alpha} = 0 \text{ and } \lim_{T \to \infty} \frac{R'_T}{T^\alpha} = 0.$$

It follows from (15) that

$$\liminf_{T \to \infty} \frac{\sum_{\mathbf{s} \in \mathcal{I}} \mathbb{E}[N_T(\mathbf{s})] D(P_\mathbf{s} \parallel P'_\mathbf{s})}{\ln T} \geq 1,$$

which sets the constraints for $\mathbb{E}[N_T(\mathbf{s})]/\ln T$ as $T \to \infty$. Since $R_T = \sum_{\mathbf{s} \in \mathcal{I}_{-\mathbf{s}_*}} \mathbb{E}[N_T(\mathbf{s})]\Delta_\mathbf{s}$, by replacing $\mathbb{E}[N_T(\mathbf{s})]/\ln T$ with $\eta_\mathbf{s}$, we finish the proof. $\square$

*Proof of Theorem 1.* We finish the proof by showing the equivalence of $\tilde{\mathbb{O}}(\mathcal{P}, \mathcal{M}_\mathcal{G})$ in Lemma 5 and $\mathbb{O}(\mathcal{P}, \mathcal{M}_\mathcal{G})$ in Theorem 1. First notice that there is no $\eta_{\mathbf{s}_*}$ in (12). Accordingly, $\eta_{\mathbf{s}_*}$ can be assigned to an arbitrarily large value to make $\sum_{\mathbf{s} \in \mathcal{I}} \eta_\mathbf{s} D(P_\mathbf{s} \parallel P'_\mathbf{s}) \geq 1$ if $P_{\mathbf{s}_*} \neq P'_{\mathbf{s}_*}$. Therefore, (13) is equivalent to

$$\sum_{\mathbf{s} \in \mathcal{I}_{-\mathbf{s}_*}} \eta_\mathbf{s} D(P_\mathbf{s} \parallel P'_\mathbf{s}) \geq 1, \forall \mathcal{P}' \in \{\mathcal{P}'' \in \mathcal{M}_\mathcal{G} \mid P_{\mathbf{s}_*} = P''_{\mathbf{s}_*}, \mathbf{s}_* \notin \mathcal{I}''_*\}.$$

It means if $\mathbf{s}_*$ is not optimal in $\mathcal{P}'$, then $\sum_{\mathbf{s} \in \mathcal{I}_{-\mathbf{s}_*}} \eta_\mathbf{s} D(P_\mathbf{s} \parallel P'_\mathbf{s}) \geq 1$. Its contrapositive statement is that if $\sum_{\mathbf{s} \in \mathcal{I}_{-\mathbf{s}_*}} \eta_\mathbf{s} D(P_\mathbf{s} \parallel P'_\mathbf{s}) < 1$, then $\mathbf{s}_*$ is also optimal in $\mathcal{P}'$. Putting it into the mathematical expression, we get

$$\max_{\mathbf{s} \in \mathcal{I}_{-\mathbf{s}_*}} \mu'_\mathbf{s} \leq \mu_*, \forall \mathcal{P}' \in \Big\{\mathcal{P}'' \in \mathcal{M}_\mathcal{G} \Big| \sum_{\mathbf{s} \in \mathcal{I}_{-\mathbf{s}_*}} \eta_\mathbf{s} D(P_\mathbf{s} \parallel P''_\mathbf{s}) < 1, P''_{\mathbf{s}_*} = P_{\mathbf{s}_*}\Big\},$$

where we use the fact $\mu_* = \mu'_*$ since $P_{\mathbf{s}_*} = P'_{\mathbf{s}_*}$. The proof is concluded. $\square$

## B  Finite time Regret Analysis of SCM-AAM in Theorem 2

### B.1  Guarantee with Forced Exploration

The SCM-AAM algorithm maintains a counter of exploration steps $N_t^e$. In expiration steps, if $\min_{\mathbf{s} \in \mathcal{I}} N_t(\mathbf{s}) \leq \epsilon N_t^e$, the SCM-AAM algorithm generates forced exploration by selecting one of the least selected intervention. Lemma 6 shows that the forced exploration mechanism ensures every intervention is persistently selected in exploration phases.

**Lemma 6.** *Let $\mathcal{R} = \{t_i\}_{i \geq 1}$ be the sequence of rounds that SCM-AAM enters the exploration phase. For any time $t_i \in \mathcal{R}$, it satisfies that*

$$\min_{\mathbf{s} \in \mathcal{I}} N_t(\mathbf{s}) \geq \frac{\epsilon i}{2}.$$

*Proof.* We first note that the following two facts are true

- $\min_{\mathbf{s} \in \mathcal{I}} N_t(\mathbf{s})$ is non-decreasing over $t$.

- If $\min_{\mathbf{s} \in \mathcal{I}} N_{t_i}(\mathbf{s}) \leq \epsilon N_{t_i}^{\mathrm{e}}$, then $\min_{\mathbf{s} \in \mathcal{I}} N_{t_{i+|\mathcal{I}|}}(\mathbf{s}) \geq \min N_{t_i}(\mathbf{s}) + 1$.

Since $N_t(\mathbf{s})$ is non-decreasing over $t$, the first statement is true. The second statement is true since otherwise, $\underline{\mathbf{S}}_t$ is selected by forced exploration at least $|\mathcal{I}|$ times without increasing $\min_{\mathbf{s} \in \mathcal{I}} N_t(\mathbf{s})$. With these two facts, we are ready to provide proof by contradiction. Suppose there exists $i \in \{1, \ldots, |\mathcal{R}|\}$ such that

$$\min_{\mathbf{s} \in \mathcal{I}} N_{t_i}(\mathbf{s}) < \frac{\epsilon i}{2}.$$

According to the first fact, we have $\forall j \geq i/2$,

$$\min_{\mathbf{s} \in \mathcal{I}} N_{t_j}(\mathbf{s}) \leq \min_{\mathbf{s} \in \mathcal{I}} N_{t_i}(\mathbf{s}) \leq \epsilon j.$$

According to SCM-AAM, $\epsilon \in (0, 1/|\mathcal{I}|)$. Then we apply the second fact,

$$\min_{\mathbf{s} \in \mathcal{I}} N_{t_i}(\mathbf{s}) \geq \frac{i - j}{|\mathcal{I}|} \geq \frac{\epsilon i}{2},$$

which creates a contradiction. $\square$

## B.2   Supporting Lemmas (Concentration Inequalities)

The following concentration inequalities are instrumental for regret analysis. For bandit with feedback drawn from arbitrary discrete distributions, a concentration bound on the empirical distribution is presented in [35, Lemma 6]. In the structured causal bandit problem with unobserved confounders, the actions space is $\mathcal{I}$, and the discrete support of feedback is $\Omega(\mathbf{V})$. At each time $t$, for each intervention $\mathbf{s} \in \mathcal{I}$, recall $\bar{P}_{\mathbf{s},t}$ is the empirical interventional distribution of $\mathbf{V}$ and $N_t(\mathbf{s})$ is the number of times the intervention $do(\mathbf{S} = \mathbf{s})$ is taken till $t$. For each intervention $\mathbf{s} \in \mathcal{I}$, the true interventional distribution is $P_{\mathbf{s}}$.

**Lemma 7** (Concentration Inequality for Information Distance [35]). *Let $\delta \geq |\mathcal{I}| (|\Omega(V)| - 1)$. Then for any $t > 0$,*

$$\mathbb{P}\left[\sum_{\mathbf{s} \in \mathcal{I}} N_t(\mathbf{s}) D(\bar{P}_{t,\mathbf{s}} \parallel P_{\mathbf{s}}) \geq \delta\right] \leq \left(\frac{\delta \lceil \delta \ln t + 1 \rceil 2e}{|\mathcal{I}| (|\Omega(V)| - 1)}\right)^{|\mathcal{I}|(|\Omega(V)|-1)} \exp(1 - \delta).$$

llowing Corollary 8 is a result of Lemma 7.

**Corollary 8.** *For any $\gamma > 0$, there exists suitably large universal constant $c$ such that*

$$\sum_{t=1}^{\infty} \mathbb{P}\left(\sum_{\mathbf{s} \in \mathcal{I}} N_t(\mathbf{s}) D(\bar{P}_{\mathbf{s},t} \parallel P_{\mathbf{s}}) \geq (1 + \gamma) \ln t\right) \leq c.$$

*Proof.* We select $t'$ such that $(1 + \gamma) \ln t' \geq |\mathcal{I}| (|\Omega(V)| - 1)$. We apply Lemma 7 to get

$$\sum_{t=1}^{\infty} \mathbb{P}\left(\sum_{\mathbf{s} \in \mathcal{I}} N_t(\mathbf{s}) D(\bar{P}_{\mathbf{s},t} \parallel P_{\mathbf{s}}) \geq (1 + \gamma) \ln t\right)$$

$$\leq t' + \sum_{t \geq t'} \mathbb{P}\left[\sum_{\mathbf{s} \in \mathcal{I}} N_t(\mathbf{s}) D(\bar{P}_{t,\mathbf{s}} \parallel P_{\mathbf{s}}) \geq (1 + \gamma) \ln t\right]$$

$$\leq t' + \sum_{t \geq t'} \frac{e}{t^{1+\gamma}} \left(\frac{(1 + \gamma) \ln t \lceil (1 + \gamma)(\ln t)^2 + 1 \rceil 2e}{|\mathcal{I}| (|\Omega(V)| - 1)}\right)^{|\mathcal{I}|(|\Omega(V)|-1)},$$

which remains finite. We conclude our proof. $\square$

Lemma 9 proposed in [9, Lemma 4.3] extends Hoeffding's inequality to provide an upper bound on the deviation of the empirical mean sampled at a stopping time. The time SCM-AAM enters the exploration phase is a stopping time, and the time each intervention is selected is also a stopping time. It will be used in multiple circumstances in regret analysis.

**Lemma 9** (Extension of Hoeffding's Inequality [9]). *Let $\{Z_t\}_{t \in \mathbb{N}_{>0}}$ be a sequence of independent random variables with values in $[0,1]$. Let $\mathcal{F}_t$ be the $\sigma$-algebra such that $\sigma(Z_1, \ldots, Z_t) \subset \mathcal{F}_t$ and the filtration $\mathcal{F} = \{\mathcal{F}_t\}_{t \in \mathbb{N}_{>0}}$. Consider $s \in \mathbb{N}$, and $T \in \mathbb{N}_{>0}$. We define $S_t = \sum_{j=1}^{t} \epsilon_j (Z_j - \mathbb{E}[Z_j])$, where $\epsilon_j \in \{0,1\}$ is a $\mathcal{F}_{j-1}$-measurable random variable. Further define $N_t = \sum_{j=1}^{t} \epsilon_j$. Define $\phi \in \{1, \ldots, T+1\}$ a $\mathcal{F}$-stopping time such that either $N_\phi \geq s$ or $\phi = T+1$. Then we have that*

$$P[S_\phi \geq N_\phi \delta] \leq \exp(-2s\delta^2).$$

*As a consequence,*

$$P[|S_\phi| \geq N_\phi \delta] \leq 2\exp(-2s\delta^2).$$

In Corollary 10, we extend Lemma 9 to bound the $L_1$ deviation of the empirical distribution.

**Corollary 10** ($L_1$ deviation of the empirical distribution). *Let $\mathcal{A}$ denote finite set $\{1, \ldots, a\}$. For two probability distribution $Q$ and $Q'$ on $\mathcal{A}$, let $\|Q' - Q\|_1 = \sum_{k=1}^{a} |Q'(k) - Q(k)|$. Let $X_t \in \mathcal{A}$ be a sequence of independent random variables with common distribution $Q$. Let $\mathcal{F}_t$ be the $\sigma$-algebra such that $\sigma(X_1, \ldots, X_t) \subset \mathcal{F}_t$ and the filtration $\mathcal{F} = \{\mathcal{F}_t\}_{t \in \mathbb{N}_{>0}}$. Let $\epsilon_t \in \{0,1\}$ is a $\mathcal{F}_{t-1}$-measurable random variable. We define*

$$N_t = \sum_{j=1}^{t} \epsilon_j, \, S_t(i) = \sum_{j=1}^{t} \epsilon_j \mathbb{1}\{X_j = i\}, \text{ and } \bar{Q}_t(i) = \frac{S_t(i)}{N_t}, \forall i \in \mathcal{A}.$$

*For $s \in \mathbb{N}$, and $T \in \mathbb{N}_{>0}$, let $\phi \in \{1, \ldots, T+1\}$ be a $\mathcal{F}$-stopping time such that either $N_\phi \geq s$ or $\phi = T+1$. Then we have*

$$P\left(\left\|\bar{Q}_\phi - Q\right\|_1 \geq \delta\right) \leq (2^a - 2) \exp\left(\frac{-s\delta^2}{2}\right).$$

*Proof.* It is known that for any distribution $Q'$ on $\mathcal{A}$,

$$\|Q' - Q\|_1 = 2 \max_{A \subseteq \mathcal{A}} (Q'(A) - Q(A)).$$

Then we apply a union bound to get

$$P\left(\left\|\bar{Q}_\phi - Q\right\|_1 \geq \delta\right) \leq \sum_{A \subseteq \mathcal{A}} P\left(\bar{Q}_\phi(A) - Q(A) \geq \frac{\delta}{2}\right)$$

$$\leq \sum_{A \subseteq \mathcal{A}: A \neq \mathcal{A} \text{ or } \emptyset} P\left(\bar{Q}_\phi(A) - Q(A) \geq \frac{\delta}{2}\right)$$

$$\leq (2^a - 2) \exp\left(\frac{-s\delta^2}{2}\right),$$

which concludes the proof. $\square$

### B.3 Supporting Lemmas (Continuity of Upper Confidence Bound)

For a constrained optimization problem, Berge's maximum theorem imposes additional restrictions on the objective function and constraint set to guarantee that the problem's solution varies smoothly with parameters. Specifically, it studies the following optimization problem

$$V(\theta) = \max_{x \in G(\theta)} F(x, \theta),$$

where $F : X \times \Theta \to \mathbb{R}$ the objective function, and $G(\theta)$ is the set of feasible values for $X$ given $\theta$ defined by the multi-valued correspondence $G : \Theta \rightrightarrows X$. We assume that $G(\theta) \subseteq X$ is compact for all $\theta \in \Theta$.

**Definition 4.** *The compact valued correspondence $G$ is continuous at $\theta \in \Omega(\theta)$ if it is both upper and lower hemicontinuous at $\theta$. The upper and lower hemicontinuous can be verified as below.*

- Upper hemicontinuous: *the correspondence $G$ is upper hemicontinuous at $\theta \in \Theta$ if $G(\theta)$ is nonempty and if, for every sequence $\{\theta_j\}$ with $\theta_j \to \theta$ and every sequence $\{x_j\}$ with $x_j \in G(\theta_j)$ for all $j$, there exists a convergent subsequence $\{x_{j_k}\}$ such that $x_{j_k} \to x \in G(\theta)$.*

- Lower hemicontinuous: *the correspondence $G$ is lower hemicontinuous at $\theta \in \Theta$ if $G(\theta)$ is nonempty and if, for every $x \in G(\theta)$ and every sequence $\{\theta_j\}$ such that $\theta_j \to \theta$, there is a $J \geq 1$ and a sequence $\{x_j\}$ such that $x_j \in G(\theta_j)$ for all $j \geq J$ and $x_j \to x$.*

*The correspondence $G$ is continuous if it is continuous at every $\theta \in \Omega(\theta)$.*

**Lemma 11** (Berge's Maximum Theorem [16, Ch. 17.31]). *Let $X \subseteq \mathbb{R}^n$ and $\Theta \subseteq \mathbb{R}^m$. Let $F : X \times \Theta \to \mathbb{R}$ be a continuous function, and let $G : \Theta \to X$ be a compact valued and continuous correspondence. Then the maximum value function*

$$V(\theta) = \max_{x \in G(\theta)} F(x, \theta)$$

*is well-defined and continuous, and the optimal correspondence*

$$x^*(\theta) = \{x \in G(\theta) \mid F(x, \theta)\}$$

*is nonempty, compact valued, and upper hemicontinuous.*

We want to leverage Berge's maximum theorem to prove the UCB index $\tilde{\mu}_{\mathbf{s}}(\mathcal{C}_{\mathbf{s}}(\eta_{\mathbf{s}}, \mathcal{P})) = \sum_{\mathbf{v} \in \Omega(\mathbf{V})} v_n \max_{\mathcal{P}' \in \mathcal{C}_{\mathbf{s}}(\eta_{\mathbf{s}}, \mathcal{P})} P'_{\mathbf{s}}(\mathbf{v})$ is continuous with respect to $\eta_{\mathbf{s}}$ and $\mathcal{P}$. Recall in Theorem 2, for given interventional distribution tuple $\mathcal{P} \in \mathcal{M}_{\mathcal{G}}$ and $\eta_{\mathbf{s}} > 0$, $\mathcal{C}_{\mathbf{s}}(\eta_{\mathbf{s}}, \mathcal{P})$ is the set of feasible interventional distribution tuples $\mathcal{P}'$, and it is defined as

$$\mathcal{C}_{\mathbf{s}}(\eta_{\mathbf{s}}, \mathcal{P}) = \left\{ \mathcal{P}' \in \mathcal{M}_{\mathcal{G}} \;\middle|\; \eta_{\mathbf{s}} D(P_{\mathbf{s}} \parallel P'_{\mathbf{s}}) + \epsilon \eta_{\mathbf{s}}/2 \sum_{\mathbf{x} \in \mathcal{I}_{-\mathbf{s}}} D(P_{\mathbf{x}} \parallel P'_{\mathbf{x}}) \leq 1 \right\}.$$

**Lemma 12.** *For any $\mathbf{s} \in \mathcal{I}$, the correspondence $\mathcal{C}_{\mathbf{s}} : \mathbb{R}_{\geq 0} \times \mathcal{M}_{\mathcal{G}} \rightrightarrows \mathcal{M}_{\mathcal{G}}$ is a compacted valued continuous correspondence.*

*Proof.* The space of interventional distribution tuples $\mathcal{M}_{\mathcal{G}}$ is compacted as has been explained below Definition 2. Besides, for any $\mathcal{P} \in \mathcal{M}_{\mathcal{G}}$ and $\eta_{\mathbf{s}} \geq 0$, the constrain $\eta_{\mathbf{s}} D(P_{\mathbf{s}} \parallel P'_{\mathbf{s}}) + \epsilon \eta_{\mathbf{s}}/2 \sum_{\mathbf{x} \in \mathcal{I}_{-\mathbf{s}}} D(P_{\mathbf{x}} \parallel P'_{\mathbf{x}}) \leq 1$ defines a closed compact set for $\mathcal{P}' = \langle P'_{\mathbf{s}} \rangle_{\mathbf{s} \in \mathcal{I}}$. Since the intersection of two compact sets is compact if one of them is also closed, $\mathcal{C}_{\mathbf{s}}(\eta_{\mathbf{s}}, \mathcal{P})$ is compact. It remains to show $\mathcal{C}_{\mathbf{s}}$ is both upper hemicontinuous and lower hemicontinuous.

**Upper hemicontinuous:** Fix arbituary $\eta_{\mathbf{s}} \geq 0$ and $\mathcal{P} \in \mathcal{M}_{\mathcal{G}}$, and let $\{(\eta_{\mathbf{s},i}, \mathcal{P}_i)\}_{i \in \mathbb{N}_{>0}}$ be a sequence such that $\eta_{\mathbf{s},i} \geq 0$, $\mathcal{P}_i \in \mathcal{M}_{\mathcal{G}}$, and $(\eta_{\mathbf{s},i}, \mathcal{P}_i) \to (\eta_{\mathbf{s}}, \mathcal{P})$. Notice that each $\mathcal{C}_{\mathbf{s}}(\eta_{\mathbf{s},i}, \mathcal{P}_i)$ is non-empty since $\mathcal{P}_i \in \mathcal{C}_{\mathbf{s}}(\eta_{\mathbf{s},i}, \mathcal{P}_i)$. Therefore, there is a sequence $\{\mathcal{P}'_i\}_{i \in \mathbb{N}_{>0}}$ such that $\mathcal{P}'_i \in \mathcal{C}_{\mathbf{s}}(\eta_{\mathbf{s},i}, \mathcal{P}_i)$. Since $\eta_{\mathbf{s},i} \to \eta_{\mathbf{s}}$, there exists a closed and bounded set $\Theta \subset \mathbb{R}_{\geq 0}$ such that $\eta_{\mathbf{s}} \in \Theta$, and for some $N \geq 1$, $\eta_{\mathbf{s},i} \in \Theta$ for all $i \geq N$. Moreover, $\mathcal{P}_i$ and $\mathcal{P}'_i$ are also bounded since every interventional probability tuple in $\mathcal{M}_{\mathcal{G}}$ is bounded. Since each $(\eta_{\mathbf{s},i}, \mathcal{P}_i, \mathcal{P}'_i,)$ lies in a closed and bounded subset for $i \geq N$, by Bolzano-Weierstrass theorem, the sequence $\{(\eta_{\mathbf{s},i}, \mathcal{P}_i, \mathcal{P}'_i,)\}_{i \in \mathbb{N}_{>0}}$ has a convergent subsequence $\{(\eta_{\mathbf{s},i_k}, \mathcal{P}_{i_k}, \mathcal{P}'_{i_k})\}_{k \in \mathbb{N}_{>0}}$ with limit point $(\eta_{\mathbf{s}}, \mathcal{P}, \mathcal{P}')$. Since each element of this sequence satisfies

$$h(\eta_{\mathbf{s},i_k}, \mathcal{P}_{i_k}, \mathcal{P}'_{i_k}) := \eta_{\mathbf{s},i_k} D(P_{\mathbf{s},i_k} \parallel P'_{\mathbf{s},i_k}) + \epsilon \eta_{\mathbf{s},i_k}/2 \sum_{\mathbf{x} \in \mathcal{I}_{-\mathbf{s}}} D(P_{\mathbf{x},i_k} \parallel P'_{\mathbf{x},i_k}) \leq 1,$$

the inequality holds at the limit point $(\eta_{\mathbf{s}}, \mathcal{P}')$, i.e.,

$$h(\eta_{\mathbf{s}}, \mathcal{P}, \mathcal{P}') = \eta_{\mathbf{s}} D(P_{\mathbf{s}} \parallel P'_{\mathbf{s}}) + \epsilon \eta_{\mathbf{s}}/2 \sum_{\mathbf{x} \in \mathcal{I}_{-\mathbf{s}}} D(P_{\mathbf{x}} \parallel P'_{\mathbf{x}}) \leq 1,$$

due to that $h$ is a continuous function. We get $\mathcal{C}_{\mathbf{s}}$ is upper hemicontinuous.

**Lower hemicontinuous:** For arbituary $\eta_{\mathbf{s}} \geq 0$ and $\mathcal{P} \in \mathcal{M}_{\mathcal{G}}$, we fix $\mathcal{P}' \in \mathcal{C}_{\mathbf{s}}(\eta_{\mathbf{s}}, \mathcal{P})$. Let $\{(\eta_{\mathbf{s},i}, \mathcal{P}_i)\}_{i \in \mathbb{N}_{>0}}$ be a sequence such that $\eta_{\mathbf{s},i} \geq 0$, $\mathcal{P}_i = \langle P_{\mathbf{s},i} \rangle_{\mathbf{s} \in \mathcal{I}} \in \mathcal{M}_{\mathcal{G}}$, and $(\eta_{\mathbf{s},i}, \mathcal{P}_i) \to (\eta_{\mathbf{s}}, \mathcal{P})$. We show there exists sequence $\{\mathcal{P}'_i\}_{i \in \mathbb{N}_{>0}}$ such that $\mathcal{P}'_i \in \mathcal{C}_{\mathbf{s}}(\eta_{\mathbf{s},i}, \mathcal{P}_i)$ for all $i$, and $\mathcal{P}'_i \to \mathcal{P}'$.

Toward this end, for each $(\eta_{\mathbf{s},i}, \mathcal{P}_i)$, if $h(\eta_{\mathbf{s},i}, \mathcal{P}_i, \mathcal{P}') \leq 1$, we set $\mathcal{P}'_i = \mathcal{P}'$ so that $\mathcal{P}'_i \in \mathcal{C}_{\mathbf{s}}(\eta_{\mathbf{s},i}, \mathcal{P}_i)$. If $c_i := h(\eta_{\mathbf{s},i}, \mathcal{P}_i, \mathcal{P}') > 1$, we construct $\mathcal{P}'_i = \langle P'_{\mathbf{s},i} \rangle_{\mathbf{s} \in \mathcal{I}}$ as the following. With Definition 2, there exist $\{\mathbf{p}^i_1, \ldots, \mathbf{p}^i_{n_c(\mathcal{G})}\}$ and $\{\mathbf{p}'_1, \ldots, \mathbf{p}'_{n_c(\mathcal{G})}\}$ associated with the interventional distribution tuple $\mathcal{P}_i$ and $\mathcal{P}'$ respectively. Let $\mathcal{P}'_i$ be associated with

$$\left\{ \frac{c_i - 1}{c_i} \mathbf{p}^i_1 + \frac{1}{c_i} \mathbf{p}'_1, \ldots, \frac{c_i - 1}{c_i} \mathbf{p}^i_{n_c(\mathcal{G})} + \frac{1}{c_i} \mathbf{p}'_{n_c(\mathcal{G})} \right\}, \tag{16}$$

so that $\mathcal{P}'_i \in \mathcal{M}_{\mathcal{G}}$. For each $\mathbf{s} \in \mathcal{I}$ and $\mathbf{v} \in \Omega(\mathbf{V})$, we have

$$P_{\mathbf{s},i}(\mathbf{v}) \ln \frac{P_{\mathbf{s},i}(\mathbf{v})}{P'_{\mathbf{s},i}(\mathbf{v})}$$

$$= P_{\mathbf{s},i}(\mathbf{v}) \ln \frac{\prod_{j=1}^{n_c(\mathcal{G})} b^\top_{\mathcal{G},\mathbf{s}[\mathbf{V}^j]}(\mathbf{v}^j, \mathsf{pa}_{\mathbf{V}^j}) \mathbf{p}^i_j}{\prod_{j=1}^{n_c(\mathcal{G})} b^\top_{\mathcal{G},\mathbf{s}[\mathbf{V}^j]}(\mathbf{v}^j, \mathsf{pa}_{\mathbf{V}^j}) \left[ \frac{c_i - 1}{c_i} \mathbf{p}^i_j + \frac{1}{c_i} \mathbf{p}'_j \right]}$$

$$= \prod_{j=1}^{n_c(\mathcal{G})} b^\top_{\mathcal{G},\mathbf{s}[\mathbf{V}^j]}(\mathbf{v}^j, \mathsf{pa}_{\mathbf{V}^j}) \mathbf{p}^i_j \sum_{k=1}^{n_c(\mathcal{G})} \ln \frac{b^\top_{\mathcal{G},\mathbf{s}[\mathbf{V}^k]}(\mathbf{v}^k, \mathsf{pa}_{\mathbf{V}^k}) \mathbf{p}^i_k}{b^\top_{\mathcal{G},\mathbf{s}[\mathbf{V}^k]}(\mathbf{v}^k, \mathsf{pa}_{\mathbf{V}^k}) \left[ \frac{c_1 - 1}{c_i} \mathbf{p}^i_k + \frac{1}{c_i} \mathbf{p}'_k \right]}$$

$$= \sum_{k=1}^{n_c(\mathcal{G})} \prod_{j \neq k} b^\top_{\mathcal{G},\mathbf{s}[\mathbf{V}^j]}(\mathbf{v}^j, \mathsf{pa}_{\mathbf{V}^j}) \mathbf{p}^i_j \underbrace{b^\top_{\mathcal{G},\mathbf{s}[\mathbf{V}^k]}(\mathbf{v}^k, \mathsf{pa}_{\mathbf{V}^k}) \mathbf{p}^i_k \ln \frac{b^\top_{\mathcal{G},\mathbf{s}[\mathbf{V}^k]}(\mathbf{v}^k, \mathsf{pa}_{\mathbf{V}^k}) \mathbf{p}^i_k}{b^\top_{\mathcal{G},\mathbf{s}[\mathbf{V}^k]}(\mathbf{v}^k, \mathsf{pa}_{\mathbf{V}^k}) \left[ \frac{c_i - 1}{c_i} \mathbf{p}^i_k + \frac{1}{c_i} \mathbf{p}'_k \right]}}_{(a)}$$

$$\leq \frac{1}{c_i} \sum_{k=1}^{n_c(\mathcal{G})} \prod_{j=1}^{n_c(\mathcal{G})} b^\top_{\mathcal{G},\mathbf{s}[\mathbf{V}^j]}(\mathbf{v}^j, \mathsf{pa}_{\mathbf{V}^j}) \mathbf{p}^i_j \ln \frac{b^\top_{\mathcal{G},\mathbf{s}[\mathbf{V}^k]}(\mathbf{v}^k, \mathsf{pa}_{\mathbf{V}^k}) \mathbf{p}^i_k}{b^\top_{\mathcal{G},\mathbf{s}[\mathbf{V}^k]}(\mathbf{v}^k, \mathsf{pa}_{\mathbf{V}^k}) \mathbf{p}'_k} = \frac{1}{c_i} P_{\mathbf{s}}(\mathbf{v}) \ln \frac{P_{\mathbf{s}}(\mathbf{v})}{P'_{\mathbf{s}}(\mathbf{v})},$$

where we apply the log sum inequality to $(a)$. Thus, for any $\mathbf{s} \in \mathcal{I}$, we have

$$D(P_{\mathbf{s},i} \parallel P'_{\mathbf{s},i}) = \sum_{\mathbf{v} \in \Omega(\mathbf{V})} P_{\mathbf{s}}(\mathbf{v}) \ln \frac{P_{\mathbf{s}}(\mathbf{v})}{P'_{\mathbf{s},i}(\mathbf{v})} \leq \frac{1}{c_i} D(P_{\mathbf{s}} \parallel P'_{\mathbf{s}}).$$

As a result, $h(\eta_{\mathbf{s},i}, \mathcal{P}_i, \mathcal{P}'_i) \leq \frac{1}{c_i} h(\eta_{\mathbf{s},i}, \mathcal{P}_i, \mathcal{P}') = 1$, which means $\mathcal{P}'_i \in \mathcal{C}_{\mathbf{s}}(\eta_{\mathbf{s},i}, \mathcal{P}_i)$.

It remains to show $\mathcal{P}'_i \to \mathcal{P}'$ by construction. If $h(\eta_{\mathbf{s}}, \mathcal{P}, \mathcal{P}') < 1$, in the sequence $\{(\eta_{\mathbf{s},i}, \mathcal{P}_i)\}_{i \in \mathbb{N}_{>0}}$, there exist finite instances such that $h(\eta_{\mathbf{s},i}, \mathcal{P}_i, \mathcal{P}') > 1$. Otherwise, $h(\eta_{\mathbf{s}}, \mathcal{P}, \mathcal{P}') = 1$ indicates $c_i \to 1$, resulting in $(16) \to \{\mathbf{p}'_1, \ldots, \mathbf{p}'_{n_c(\mathcal{G})}\}$. Therefore, $\mathcal{P}'_i \to \mathcal{P}'$ and $\mathcal{C}_{\mathbf{s}}$ is lower hemicontinuous. $\square$

With correspondence $\mathcal{C}_{\mathbf{s}}$ being continuous, Lemma 13 gives a continuous property of $\tilde{\mu}_{\mathbf{s}}(\mathcal{C}_{\mathbf{s}}(\eta_{\mathbf{s}}, \mathcal{P}))$.

**Lemma 13.** *For any $\mathcal{P} \in \mathcal{M}_{\mathcal{G}}$ and for any any $\mathbf{s} \in \mathcal{I}$, the UCB index $\tilde{\mu}_{\mathbf{s}}(\mathcal{C}_{\mathbf{s}}(\eta_{\mathbf{s}}, \mathcal{P}))$ is a is continuous monotonically nonincreasing function of $\eta_{\mathbf{s}} \geq 0$.*

*Proof.* Since $\mathcal{C}_{\mathbf{s}}$ is a compacted valued continuous correspondence, we apply Berge's maximum theorem in Lemma 11 to get that $\max_{\mathcal{P}' \in \mathcal{C}_{\mathbf{s}}(\eta_{\mathbf{s}}, \mathcal{P})} P'_{\mathbf{s}}(\mathbf{v})$ is continuous with $\eta_{\mathbf{s}}$. So $\tilde{\mu}_{\mathbf{s}}(\mathcal{C}_{\mathbf{s}}(\eta_{\mathbf{s}}, \mathcal{P})) = \sum_{\mathbf{v} \in \Omega(\mathbf{V})} v_n \max_{\mathcal{P}' \in \mathcal{C}_{\mathbf{s}}(\eta_{\mathbf{s}}, \mathcal{P})} P'_{\mathbf{s}}(\mathbf{v})$ is also continuous with $\eta_{\mathbf{s}}$. For any $a > b > 0$, since $\mathcal{C}_{\mathbf{s}}(a, \mathcal{P}) \subseteq \mathcal{C}_{\mathbf{s}}(b, \mathcal{P})$, $\tilde{\mu}_{\mathbf{s}}(\mathcal{C}_{\mathbf{s}}(a, \mathcal{P})) \leq \tilde{\mu}_{\mathbf{s}}(\mathcal{C}_{\mathbf{s}}(b, \mathcal{P}))$. Therefore, $\tilde{\mu}_{\mathbf{s}}(\mathcal{C}_{\mathbf{s}}(\eta_{\mathbf{s}}, \mathcal{P}))$ monotonically nonincreasing with respect to $\eta_{\mathbf{s}}$. $\square$

The SCM-AAM utilizes the empirical interventional distributions to compute UCB and mean reward in exploration steps. Lemma 14 indicates that only a small ratio of exploration $\kappa$ is wasted if the gap between empirical distribution and the ground truth is as small as $\delta(\kappa)$. This fact is the cornerstone of regret analysis since the ground truth is unavailable.

**Lemma 14.** *Given a causal graph $\mathcal{G} = \langle \mathbf{V}, \mathcal{E}, \mathcal{B} \rangle$, for any interventional distribution tuple $\mathcal{P}' = \langle P'_{\mathbf{s}} \rangle_{\mathbf{s} \in \mathcal{I}} \in \mathcal{M}_{\mathcal{G}}$, let $\mu_{\mathbf{s}}(\mathcal{P}')$ be the mean reward of each intervention $\mathbf{s} \in \mathcal{I}$ according to $\mathcal{P}'$, and define $\mu_*(\mathcal{P}') = \max_{\mathbf{s} \in \mathcal{I}} \mu_{\mathbf{s}}(\mathcal{P}')$. Let $\mathcal{P} = \langle P_{\mathbf{s}} \rangle_{\mathbf{s} \in \mathcal{I}} \in \mathcal{M}_{\mathcal{G}}$ be the true interventional distribution tuple with a unique optimal intervention $\mathbf{s}_* \in \mathcal{I}$. For each $\mathbf{s} \in \mathcal{I}_{-\mathbf{s}_*}$, let*

$$\zeta_{\mathbf{s}}(\mathcal{P}) = \max_{\eta_{\mathbf{s}} \geq 0} \eta_{\mathbf{s}} \quad \text{s.t.} \quad \tilde{\mu}_{\mathbf{s}}(\mathcal{C}_{\mathbf{s}}(\eta_{\mathbf{s}}, \mathcal{P})) \geq \mu_*(\mathcal{P}), \tag{17}$$

*in which* $\mathcal{C}_{\mathbf{s}}(\eta_{\mathbf{s}}, \mathcal{P}) := \left\{ \mathcal{P}' \in \mathcal{M}_{\mathcal{G}} \mid \eta_{\mathbf{s}} D(P_{\mathbf{s}} \parallel P'_{\mathbf{s}}) + \epsilon \eta_{\mathbf{s}}/2 \sum_{\mathbf{x} \in \mathcal{I}_{-\mathbf{s}}} D(P_{\mathbf{x}} \parallel P'_{\mathbf{x}}) \leq 1 \right\}$. *For any* $\kappa > 0$, *there exists* $\delta(\kappa)$ *such that for any* $\bar{\mathcal{P}} = \langle \bar{P}_{\mathbf{s}} \rangle_{\mathbf{s} \in \mathcal{I}} \in \mathcal{M}_{\mathcal{G}}$, *if* $\forall : \mathbf{s} \in \mathcal{I} : \left\| \bar{P}_{\mathbf{s}} - P_{\mathbf{s}} \right\|_1 \leq \delta(\kappa)$,

- *Intervention* $\mathbf{s}_*$ *is also optimal according to* $\bar{\mathcal{P}}$, *i.e.,* $\mu_{\mathbf{s}_*}(\bar{\mathcal{P}}) \geq \mu_{\mathbf{s}}(\bar{\mathcal{P}}), \forall \mathbf{s} \in \mathcal{I}_{-\mathbf{s}_*}$.

- *For any* $\mathbf{s} \in \mathcal{I}_{-\mathbf{s}_*}$, $\zeta_{\mathbf{s}}(\bar{\mathcal{P}}) \leq (1 + \kappa) \zeta_{\mathbf{s}}(\mathcal{P})$.

*Proof.* For each $\mathbf{s} \in \mathcal{I}$, it can be seen that $\lim_{\eta_{\mathbf{s}} \to \infty} \tilde{\mu}_{\mathbf{s}}(\mathcal{C}_{\mathbf{s}}(\eta_{\mathbf{s}}, \mathcal{P})) = \mu_{\mathbf{s}}(\mathcal{P})$ since $\mathcal{P}'$ in $\mathcal{C}_{\mathbf{s}}(\eta_{\mathbf{s}}, \mathcal{P})$ is required to be identical to $\mathcal{P}$ as $\eta_{\mathbf{s} \to \infty}$. For each $\mathbf{s} \in \mathcal{I}_{-\mathbf{s}_*}$, if $\tilde{\mu}_{\mathbf{s}}(\mathcal{C}_{\mathbf{s}}(0, \mathcal{P})) > \mu_*(\mathcal{P})$, since $\tilde{\mu}_{\mathbf{s}}(\mathcal{C}_{\mathbf{s}}(\eta_{\mathbf{s}}, \mathcal{P}))$ is a continuous monotonically nonincreasing function of $\eta_{\mathbf{s}}$ according to Lemma 13, there exists $\eta_{\mathbf{s}} > 0$ such that $\tilde{\mu}_{\mathbf{s}}(\mathcal{C}_{\mathbf{s}}(\eta_{\mathbf{s}}, \mathcal{P})) = \mu_*(\mathcal{P})$. Setting $\eta_{\mathbf{s}}^* = \max\{\eta_{\mathbf{s}} \geq 0 \mid \tilde{\mu}_{\mathbf{s}}(\mathcal{C}_{\mathbf{s}}(\eta_{\mathbf{s}}, \mathcal{P})) = \mu_*(\mathcal{P})\}$, we have $\eta_{\mathbf{s}}^* > 0$ and $\zeta_{\mathbf{s}}(\mathcal{P}) = \eta_{\mathbf{s}}^*$. For any $\kappa > 0$, $\tilde{\mu}_{\mathbf{s}}(\mathcal{C}_{\mathbf{s}}(\sqrt{1 + \kappa}\eta_{\mathbf{s}}^*, \mathcal{P})) < \mu_*(\mathcal{P})$ due to the definition of $\eta_{\mathbf{s}}^*$. We define

$$\delta'_{\mathbf{s}}(\kappa) = 2\mu_*(\mathcal{P}) - 2\tilde{\mu}_{\mathbf{s}}(\mathcal{C}_{\mathbf{s}}(\sqrt{1 + \kappa}\eta_{\mathbf{s}}^*, \mathcal{P})) > 0. \tag{18}$$

From Lemma 12, $\mathcal{C}_{\mathbf{s}}$ is a continuous correspondence. Since $D(P_{\mathbf{s}} \parallel P'_{\mathbf{s}}) + \epsilon/2 \sum_{\mathbf{x} \in \mathcal{I}_{-\mathbf{s}}} D(P_{\mathbf{x}} \parallel P'_{\mathbf{x}})$ is a continuous function of $\mathcal{P}' = \langle P'_{\mathbf{s}} \rangle_{\mathbf{s} \in \mathcal{I}} \in \mathcal{M}_{\mathcal{G}}$, it follows from Berge's maximum theorem in Lemma 11 that

$$\max_{\mathcal{P}' \in \mathcal{C}_{\mathbf{s}}((1+\kappa)\eta_{\mathbf{s}}, \bar{\mathcal{P}})} D(\bar{P}_{\mathbf{s}} \parallel P'_{\mathbf{s}}) + \epsilon/2 \sum_{\mathbf{x} \in \mathcal{I}_{-\mathbf{s}}} D(\bar{P}_{\mathbf{x}} \parallel P'_{\mathbf{x}})$$

is a continuous function of $\bar{\mathcal{P}} = \langle \bar{P}_{\mathbf{s}} \rangle_{\mathbf{s} \in \mathcal{I}}$. Thus, there exists $\delta''_{\mathbf{s}}(\kappa) > 0$ such that if $\forall \mathbf{s} \in \mathcal{I} : \left\| \bar{P}_{\mathbf{s}} - P_{\mathbf{s}} \right\|_1 \leq \delta''_{\mathbf{s}}(\kappa)$,

$$\max_{\mathcal{P}' \in \mathcal{C}_{\mathbf{s}}((1+\kappa)\eta_{\mathbf{s}}^*, \bar{\mathcal{P}})} D(P_{\mathbf{s}} \parallel P'_{\mathbf{s}}) + \epsilon/2 \sum_{\mathbf{x} \in \mathcal{I}_{-\mathbf{s}}} D(P_{\mathbf{x}} \parallel P'_{\mathbf{x}})$$

$$\leq \sqrt{1 + \kappa} \max_{\mathcal{P}' \in \mathcal{C}_{\mathbf{s}}((1+\kappa)\eta_{\mathbf{s}}^*, \mathcal{P})} D(P_{\mathbf{s}} \parallel P'_{\mathbf{s}}) + \epsilon/2 \sum_{\mathbf{x} \in \mathcal{I}_{-\mathbf{s}}} D(P_{\mathbf{x}} \parallel P'_{\mathbf{x}}) \leq \frac{1}{\eta_{\mathbf{s}}^* \sqrt{1 + \kappa}},$$

where the second inequality is due to the definition of $\mathcal{C}_{\mathbf{s}}((1 + \kappa)\eta_{\mathbf{s}}, \mathcal{P})$. The inequality means for any $\mathcal{P}' \in \mathcal{C}_{\mathbf{s}}((1 + \kappa)\eta_{\mathbf{s}}^*, \bar{\mathcal{P}})$,

$$(\eta_{\mathbf{s}}^* \sqrt{1 + \kappa}) D(P_{\mathbf{s}} \parallel P'_{\mathbf{s}}) + (\eta_{\mathbf{s}}^* \sqrt{1 + \kappa}) \epsilon/2 \sum_{\mathbf{x} \in \mathcal{I}_{-\mathbf{s}}} D(P_{\mathbf{x}} \parallel P'_{\mathbf{x}}) \leq 1,$$

which means $\mathcal{C}_{\mathbf{s}}((1 + \kappa)\eta_{\mathbf{s}}^*, \bar{\mathcal{P}}) \subseteq \mathcal{C}_{\mathbf{s}}(\sqrt{1 + \kappa}\eta_{\mathbf{s}}^*, \mathcal{P})$. Therefore,

$$\tilde{\mu}_{\mathbf{s}}(\mathcal{C}_{\mathbf{s}}((1 + \kappa)\eta_{\mathbf{s}}^*, \bar{\mathcal{P}})) \leq \tilde{\mu}_{\mathbf{s}}(\mathcal{C}_{\mathbf{s}}(\sqrt{1 + \kappa}\eta_{\mathbf{s}}^*, \mathcal{P})). \tag{19}$$

Let $\hat{\delta}_{\mathbf{s}}(\kappa) = \min\{\delta'_{\mathbf{s}}(\kappa), \delta''_{\mathbf{s}}(\kappa)\}$, and let $\hat{\delta}(\kappa) = \min_{\mathbf{s} \in \mathcal{I}_{-\mathbf{s}_*}} \hat{\delta}_{\mathbf{s}}(\kappa)$. Then we set $\delta(\kappa) = \min\{\Delta_{\min}, \hat{\delta}(\kappa)\}$, where $\Delta_{\min} = \min_{\mathbf{s} \in \mathcal{I}_{-\mathbf{s}_*}} \mu_*(\mathcal{P}) - \mu_{\mathbf{s}}(\mathcal{P})$. Since the reward is bounded by $[0, 1]$, for any $\mathbf{s} \in \mathcal{I}$, $\left| \mu_{\mathbf{s}}(\bar{\mathcal{P}}) - \mu_{\mathbf{s}}(\mathcal{P}) \right| \leq \frac{1}{2} \left\| \bar{P}_{\mathbf{s}} - P_{\mathbf{s}} \right\|_1$. If $\forall \mathbf{s} \in \mathcal{I} : \left\| \bar{P}_{\mathbf{s}} - P_{\mathbf{s}} \right\|_1 \leq \delta(\kappa)$,

$$\mu_{\mathbf{s}}(\bar{\mathcal{P}}) - \mu_{\mathbf{s}_*}(\bar{\mathcal{P}}) \leq \mu_{\mathbf{s}}(\mathcal{P}) - \mu_{\mathbf{s}_*}(\mathcal{P}) + \delta(\kappa) \leq \mu_{\mathbf{s}}(\mathcal{P}) - \mu_{\mathbf{s}_*}(\mathcal{P}) + \Delta_{\min} \leq 0,$$

which means intervention $\mathbf{s}_*$ is also optimal according to $\bar{\mathcal{P}}$. Besides, with (18) and (19), we have

$$\tilde{\mu}_{\mathbf{s}}(\mathcal{C}_{\mathbf{s}}((1 + \kappa)\eta_{\mathbf{s}}^*, \bar{\mathcal{P}})) \leq \tilde{\mu}_{\mathbf{s}}(\mathcal{C}_{\mathbf{s}}(\sqrt{1 + \kappa}\eta_{\mathbf{s}}^*, \mathcal{P})) = \mu_*(\mathcal{P}) - \frac{\delta'_{\mathbf{s}}(\kappa)}{2}$$

$$\leq \mu_*(\mathcal{P}) - \frac{\delta(\kappa)}{2} \leq \mu_{\mathbf{s}_*}(\mathcal{P}) - \frac{1}{2} \left\| \bar{P}_{\mathbf{s}_*} + P_{\mathbf{s}_*} \right\|_1 \leq \mu_{\mathbf{s}_*}(\bar{\mathcal{P}}) = \mu_*(\bar{\mathcal{P}}).$$

Since $\tilde{\mu}_{\mathbf{s}}(\mathcal{C}_{\mathbf{s}}(\eta_{\mathbf{s}}, \bar{\mathcal{P}}))$ is continuous monotonically nonincreasing with $\eta_{\mathbf{s}}$ and $\tilde{\mu}_{\mathbf{s}}(\mathcal{C}_{\mathbf{s}}(\zeta_{\mathbf{s}}(\bar{\mathcal{P}}), \bar{\mathcal{P}})) \geq \mu_*(\bar{\mathcal{P}}) \geq \tilde{\mu}_{\mathbf{s}}(\mathcal{C}_{\mathbf{s}}((1 + \kappa)\eta_{\mathbf{s}}^*, \bar{\mathcal{P}}))$, we have

$$\zeta_{\mathbf{s}}(\bar{\mathcal{P}}) \leq (1 + \kappa)\eta_{\mathbf{s}}^* = (1 + \kappa)\zeta_{\mathbf{s}}(\mathcal{P}),$$

which concludes the proof. $\qquad \square$

## B.4 Main Proof of Theorem 2

The expected regret is defined as $R_T^{\text{SCM-AAM}} = \sum_{\mathbf{s} \in \mathcal{I}_{-\mathbf{s}_*}} \mathbb{E}[N_T(\mathbf{s})] \Delta_{\mathbf{s}}$, in which $N_T(\mathbf{s})$ is the total number of times $\mathbf{s}$ is selected by the SCM-AAM algorithm till $T$. So the proof is articulated around bounding $\mathbb{E}[N_T(\mathbf{s})]$ for each $\mathbf{s} \in \mathcal{I}_{-\mathbf{s}_*}$.

*Proof.* At each time $t$, consider the event that the true interventional distribution tuple $\mathcal{P} = \langle P_{\mathbf{s}} \rangle_{\mathbf{s} \in \mathcal{I}}$ is not contained by the confidence set $\mathcal{C}_t = \left\{ \mathcal{P}' \in \mathcal{M}_{\mathcal{G}} \;\middle|\; \sum_{\mathbf{s} \in \mathcal{I}} N_t(\mathbf{s}) D(\bar{P}_{\mathbf{s},t} \| P'_{\mathbf{s}}) \leq (1+\gamma) \ln t \right\}$. Due to Corollary 8, there exists $c_1 > 0$ irrelevant with $T$

$$\sum_{t=1}^{T} \mathbb{P}(\mathcal{P} \notin \mathcal{C}_t) \leq \sum_{t=1}^{T} \mathbb{P}\left( \sum_{\mathbf{s} \in \mathcal{I}} N_t(\mathbf{s}) D(\bar{P}_{\mathbf{s},t} \| P_{\mathbf{s}}) \geq (1+\gamma) \ln t \right) \leq c_1, \tag{20}$$

which will only result in finite regret bounded $c_1$ since reward is in $[0,1]$. So by only considering time steps with $\{\mathcal{P} \in \mathcal{C}_t\}$ to be true, we upper bound the number of times a suboptimal arm $\mathbf{s} \in \mathcal{I}_{-\mathbf{s}_*}$ is selected in the exploration and exploitation phase separately.

**Exploitation phase:** When $\{\mathcal{P} \in \mathcal{C}_t\}$ is true, we have

$$\forall \mathbf{s} \in \mathcal{I} : \tilde{\mu}_{\mathbf{s}}(\mathcal{C}_t) = \sum_{\mathbf{v} \in \Omega(\mathbf{V})} v_n \max_{\mathcal{P}' \in \mathcal{C}_t} P'_{\mathbf{s}}(\mathbf{v}) \geq \sum_{\mathbf{v} \in \Omega(\mathbf{V})} v_n P_{\mathbf{s}}(\mathbf{v}) = \mu_{\mathbf{s}}. \tag{21}$$

According to the design, SCM-AAM enters the exploitation phase at time $t$ if $\bar{\mu}_{\mathbf{S}_t, t} > \tilde{\mu}_{\mathbf{s}}(\mathcal{C}_t), \forall \mathbf{s} \in \mathcal{I} \setminus \{\mathbf{S}_t\}$. It follows from (21), if $\mathbf{S}_t \neq \mathbf{s}_*$ and $\mathcal{P} \in \mathcal{C}_t$,

$$\bar{\mu}_{\mathbf{S}_t, t} > \tilde{\mu}_{\mathbf{s}_*}(\mathcal{C}_t) \geq \mu_*.$$

Thus, we can bound the regret from exploitation steps as the following

$$\sum_{t=1}^{T} \sum_{\mathbf{s} \in \mathcal{I} \setminus \{\mathbf{s}_*\}} \Delta_{\mathbf{s}} \mathbb{1}\{\mathbf{S}_t = \mathbf{s}, \mathcal{P} \in \mathcal{C}_t\} \leq \sum_{t=1}^{T} \sum_{\mathbf{s} \in \mathcal{I}/\{\mathbf{s}_*\}} \Delta_{\mathbf{s}} \mathbb{1}\{\mathbf{S}_t = \mathbf{s}, \bar{\mu}_{\mathbf{s},t} > \mu_*.\} \tag{22}$$

Let $\{t_{\mathbf{s},i}\}_{i \geq 1}$ be the sequence of rounds such that $\mathbf{s}$ is selected by exploitation steps, namely $\mathbf{S}_{t_{\mathbf{s},i}} = \mathbf{s}$. Considering the warmup step that selects each intervention once, at each $t_{\mathbf{s},i}$, we have $N_{t_{\mathbf{s},i}}(\mathbf{s}) \geq i$. Since each $t_{\mathbf{s},i}$ is a stopping time, by applying Lemma 9, we bound the expectation of (22) as

$$\mathbb{E}\left[ \sum_{t=1}^{T} \sum_{\mathbf{s} \in \mathcal{I}_{-\mathbf{s}_*}} \Delta_{\mathbf{s}} \mathbb{1}\{\mathbf{S}_t = \mathbf{s}, \mathcal{P} \in \mathcal{C}_t\} \right] \leq \sum_{\mathbf{s} \in \mathcal{I}_{-\mathbf{s}_*}} \Delta_{\mathbf{s}} \sum_{i \geq 1} P(\bar{\mu}_{\mathbf{s}, t_{\mathbf{s},i}} - \mu_{\mathbf{s}} > \Delta_{\mathbf{s}}, t_{\mathbf{s},i} \leq T)$$

$$\leq \sum_{\mathbf{s} \in \mathcal{I}_{-\mathbf{s}_*}} \Delta_{\mathbf{s}} \sum_{i \geq 1} \exp(-2i\Delta_{\mathbf{s}}^2) \leq \sum_{\mathbf{s} \in \mathcal{I}_{-\mathbf{s}_*}} \frac{1}{2\Delta_{\mathbf{s}}} := c_2. \tag{23}$$

**Exploration phase with inaccurate $\bar{\mathcal{P}}_t$:** We define event $E_t$ that $\bar{P}_t$ is estimated with sufficient accuracy,

$$E_t = \left\{ \forall \mathbf{s} \in \mathcal{I} : \left\| \bar{P}_{t,\mathbf{s}} - P_{\mathbf{s}} \right\|_1 \leq \delta(\kappa) \right\}, \tag{24}$$

where $\delta(\kappa) > 0$ is defined in Lemma 14. Let $\mathcal{R} = \{t_i\}_{i \geq 1}$ be the sequence of rounds that SCM-AAM enters the exploration phase, and each $t_i$ is a stopping time. Then we have

$$\sum_{t=1}^{T} \mathbb{1}\{t \in \mathcal{R}, E_t^{\complement}\} \leq \sum_{i \geq 1} \mathbb{1}\{E_{t_i}^{\complement}, t_i \leq T\}.$$

From Lemma 6, at $t_i$, we have $\forall \mathbf{s} \in \mathcal{I} : N_t(\mathbf{s}) \geq \epsilon i / 2$. With Corollary 10, we apply a union bound

$$\mathbb{E}\left[ \sum_{t=1}^{T} \mathbb{1}\{t \in \mathcal{R}, E_t^{\complement}\} \right] \leq \sum_{\mathbf{s} \in \mathcal{I}} \sum_{i \geq 1} P(E_{t_i}^{\complement}, t_i \leq T) \leq \sum_{i \geq 1} P\left( \left\| \bar{P}_{t_i, \mathbf{s}} - P_{\mathbf{s}} \right\|_1 > \delta(\kappa), t_i \leq T \right)$$

$$\leq \sum_{\mathbf{s} \in \mathcal{I}} \sum_{i \geq 1} (2^{|\Omega(\mathbf{V})|} - 2) \exp\left( \frac{-\epsilon i \delta(\kappa)^2}{4} \right) = \frac{(2^{|\Omega(\mathbf{V})|+2} - 8) |\mathcal{I}|}{\epsilon \delta(\kappa)^2} = c_3.$$

$$\tag{25}$$

**Greedy approximation exploration with accurate** $\bar{\mathcal{P}}_t$**:** For each $\mathbf{s} \in \mathcal{I}$, let $N_t^{\mathrm{e}}(\mathbf{s})$ be the total number of times $\mathbf{s}$ is selected in exploration phases till $t$. Within time steps such that both $E_t$ defined in (24) and $\{\mathcal{P} \in \mathcal{C}_t\}$ are true, let the last time $\mathbf{s} \in \mathcal{I}_{-\mathbf{s}_*}$ being selected by exploration with greedy approximation be

$$\bar{t}_\mathbf{s} = \max\{t \in \mathcal{R} \mid \overline{\mathbf{S}}_t = \mathbf{s}, E_t, \mathcal{P} \in \mathcal{C}_t\}$$

is true. With Lemma 6, we have $\forall \mathbf{x} \in \mathcal{I}_{-\mathbf{s}} : N_{\bar{t}_\mathbf{s}}(\mathbf{x}) \geq \epsilon N_{\bar{t}_\mathbf{s}}^{\mathrm{e}}(\mathbf{s})/2$, so that

$$\frac{\sum_{\mathbf{x} \in \mathcal{I}} N_{\bar{t}_\mathbf{x}}(\mathbf{x}) D(\bar{P}_{\mathbf{x},\bar{t}_\mathbf{s}} \parallel P_\mathbf{x}')}{(1+\gamma) \ln \bar{t}_\mathbf{s}} \geq \frac{\epsilon N_{\bar{t}_\mathbf{s}}^{\mathrm{e}}(\mathbf{s}) \sum_{\mathbf{x} \in \mathcal{I}_{-\mathbf{s}}} D(\bar{P}_{\mathbf{x},\bar{t}_\mathbf{s}} \parallel P_\mathbf{x}')}{2(1+\gamma) \ln \bar{t}_\mathbf{s}} + \frac{N_{\bar{t}_\mathbf{s}}^{\mathrm{e}}(\mathbf{s}) D(\bar{P}_{\mathbf{s},\bar{t}_\mathbf{s}} \parallel P_\mathbf{s}')}{(1+\gamma) \ln \bar{t}_\mathbf{s}}. \quad (26)$$

Recall the definition of $\mathcal{C}_t$ and $\mathcal{C}_\mathbf{s}(\eta_\mathbf{s}, \bar{\mathcal{P}})$:

$$\mathcal{C}_t = \left\{ \mathcal{P}' \in \mathcal{M}_\mathcal{G} \mid \sum_{\mathbf{x} \in \mathcal{I}} N_t(\mathbf{x}) D(\bar{P}_{\mathbf{x},t} \parallel P_\mathbf{x}') \leq (1+\gamma) \ln t \right\},$$

$$\mathcal{C}_\mathbf{s}(\eta_\mathbf{s}, \bar{\mathcal{P}}_t) = \left\{ \mathcal{P}' \in \mathcal{M}_\mathcal{G} \mid \eta_\mathbf{s} D(\bar{P}_{\mathbf{s},t} \parallel P_\mathbf{s}') + \epsilon \eta_\mathbf{s}/2 \sum_{\mathbf{x} \in \mathcal{I}_{-\mathbf{s}}} D(\bar{P}_{\mathbf{x},t} \parallel P_\mathbf{x}') \leq 1 \right\}.$$

With (26), we have $\mathcal{C}_{\bar{t}_\mathbf{s}} \subseteq \mathcal{C}_\mathbf{s}\left( \frac{N_{\bar{t}_\mathbf{s}}^{\mathrm{e}}(\mathbf{s})}{(1+\gamma) \ln \bar{t}_\mathbf{s}}, \bar{\mathcal{P}}_{\bar{t}_\mathbf{s}} \right)$, and as a result,

$$\tilde{\mu}_\mathbf{s}\left( \mathcal{C}_\mathbf{s}\left( \frac{N_{\bar{t}_\mathbf{s}}^{\mathrm{e}}(\mathbf{s})}{(1+\gamma) \ln \bar{t}_\mathbf{s}}, \bar{\mathcal{P}}_t \right) \right) \geq \tilde{\mu}_\mathbf{s}(\mathcal{C}_t). \quad (27)$$

By the algorithm design, we have $\tilde{\mu}_\mathbf{s}(\mathcal{C}_{\bar{t}_\mathbf{s}}) \geq \bar{\mu}_{\mathbf{s}_*,t}$, since otherwise SCM-AAM does not enter exploration phase. Therefore, it follows from (27) that

$$\tilde{\mu}_\mathbf{s}\left( \mathcal{C}_\mathbf{s}\left( \frac{N_{\bar{t}_\mathbf{s}}^{\mathrm{e}}(\mathbf{s})}{(1+\gamma) \ln \bar{t}_\mathbf{s}}, \bar{\mathcal{P}}_{\bar{t}_\mathbf{s}} \right) \right) \geq \bar{\mu}_{\mathbf{s}_*,\bar{t}_\mathbf{s}} = \mu_*(\bar{\mathcal{P}}_{\bar{t}_\mathbf{s}}),$$

where the inequality holds since when $E_{\bar{t}_\mathbf{s}}$ happens, Lemma 14 shows $\mathbf{s}_*$ is also optimal according to $\bar{\mathcal{P}}_{\bar{t}_\mathbf{s}}$. With Lemma 13, $\tilde{\mu}_\mathbf{s}(\mathcal{C}_\mathbf{s}(\eta_\mathbf{s}, \bar{\mathcal{P}}_{\bar{t}_\mathbf{s}}))$ is a continuous nonincreasing function of $\eta_\mathbf{s}$, then we have

$$N_{\bar{t}_\mathbf{s}}^{\mathrm{e}}(\mathbf{s}) \leq \zeta_\mathbf{s}(\bar{\mathcal{P}}_{\bar{t}_\mathbf{s}})(1+\gamma) \ln \bar{t}_\mathbf{s} \leq \zeta_\mathbf{s}(\bar{\mathcal{P}}_{\bar{t}_\mathbf{s}})(1+\gamma) \ln T,$$

where $\zeta_\mathbf{s}(\bar{\mathcal{P}}_{\bar{t}_\mathbf{s}})$ defined in (17) is the maximum value of $\eta_\mathbf{s}$ such that $\tilde{\mu}_\mathbf{s}(\mathcal{C}_\mathbf{s}(\eta_\mathbf{s}, \bar{\mathcal{P}}_{\bar{t}_\mathbf{s}})) \geq \mu_*(\bar{\mathcal{P}}_{\bar{t}_\mathbf{s}})$. Together with Lemma 14, we have

$$\sum_{t=1}^{T} \mathbb{1}\{\overline{\mathcal{S}}_t = \mathbf{s}, E_t, \mathcal{P} \in \mathcal{C}_t\} = N_{\bar{t}_\mathbf{s}}^{\mathrm{e}}(\mathbf{s}) \leq \zeta_\mathbf{s}(\mathcal{P})(1+\kappa)(1+\gamma) \ln T. \quad (28)$$

**Forced Exploration:** By slightly abusing the notation, at time $t$, we denote by $\{\overline{\mathbf{S}}_t\}$ as the greedy approximation exploration event and by $\{\underline{\mathbf{S}}_t\}$ as the force exploration event. Let the last time $\mathbf{s} \in \mathcal{I}$ be selected by forced exploration be $\underline{t}_\mathbf{s} := \max\{t \in \mathcal{R} \mid \underline{\mathbf{S}}_t = \mathbf{s}\}$. When SCM-AAM force exploration at $t$, we have $N_t(\underline{\mathbf{S}}_t) \leq \epsilon N_t^{\mathrm{e}}$, resulting in

$$\sum_{t=1}^{T} \mathbb{1}\{\underline{\mathbf{S}}_t\} \leq \sum_{\mathbf{s} \in \mathcal{I}} \sum_{t=1}^{T} \mathbb{1}\{\underline{\mathbf{S}}_t = \mathbf{s}\} \leq \sum_{\mathbf{s} \in \mathcal{I}} N_{\underline{t}_\mathbf{s}}(\mathbf{s}) \leq \sum_{\mathbf{s} \in \mathcal{I}} \epsilon N_{\underline{t}_\mathbf{s}}^{\mathrm{e}} \leq \epsilon |\mathcal{I}| N_T^{\mathrm{e}}. \quad (29)$$

We want to provide an upper bound $N_T^{\mathrm{e}}$, which is

$$N_T^{\mathrm{e}} = \sum_{t=1}^{T} \mathbb{1}\{t \in \mathcal{R}\} \leq \sum_{t=1}^{T} \left( \mathbb{1}\{\underline{\mathbf{S}}_t\} + \mathbb{1}\{\overline{\mathbf{S}}_t, E_t, \mathcal{P} \in \mathcal{C}_t\} + \mathbb{1}\{t \in \mathcal{R}, E_t^{\complement}\} + \mathbb{1}\{\mathcal{P} \in \mathcal{C}_t^{\complement}\} \right), \quad (30)$$

Putting together (29) and (30), we get

$$\sum_{t=1}^{T} \mathbb{1}\{\underline{\mathbf{S}}_t\} \leq \frac{\epsilon |\mathcal{I}|}{1 - \epsilon |\mathcal{I}|} \sum_{t=1}^{T} \left( \mathbb{1}\{t \in \mathcal{R}, \overline{\mathbf{S}}_t, E_t, \mathcal{P} \in \mathcal{C}_t\} + \mathbb{1}\{t \in \mathcal{R}, E_t^{\complement}\} + \mathbb{1}\{\mathcal{P} \in \mathcal{C}_t^{\complement}\} \right).$$

Also with (28), we have

$$\sum_{t=1}^{T} \mathbb{1}\{t \in \mathcal{R}, \overline{\mathbf{S}}_t, E_t, \mathcal{P} \in \mathcal{C}_t\} = \sum_{\mathbf{s} \in \mathcal{I}} \sum_{t=1}^{T} \mathbb{1}\{t \in \mathcal{R}, \overline{\mathbf{S}}_t = \mathbf{s}, E_t, \mathcal{P} \in \mathcal{C}_t\}$$

$$\leq \sum_{\mathbf{s} \in \mathcal{I}} \zeta_{\mathbf{s}}(\overline{\mathcal{P}}_{\bar{t}_{\mathbf{s}}})(1+\kappa)(1+\gamma)\ln T.$$

Then, the expected number of forced explorations can be bound as the following,

$$\mathbb{E}\left[\sum_{t=1}^{T} \mathbb{1}\{\underline{\mathbf{S}}_t\}\right] \leq \frac{\epsilon |\mathcal{I}|}{1 - \epsilon |\mathcal{I}|}\left[c_1 + c_3 + (1+\kappa)(1+\gamma)\sum_{\mathbf{s} \in \mathcal{I}} \zeta_{\mathbf{s}}(\overline{\mathcal{P}}_{\bar{t}_{\mathbf{s}}})\ln T\right], \tag{31}$$

where $C_1$ and $C_3$ are defined in (20) and (25) respectively.

**Summary:** With reward $V_n \in [0,1]$, we decomposed the expected regret as

$$R_T^{\text{SCM-AAM}}(\mathcal{P}) \leq \mathbb{E}\left[\sum_{t=1}^{T}\sum_{\mathbf{s} \in \mathcal{I}_{-\mathbf{s}_*}} \Delta_{\mathbf{s}} \mathbb{1}\{\mathbf{S}_t = \mathbf{s}, \mathcal{P} \in \mathcal{C}_t\}\right] + \sum_{\mathbf{s} \in \mathcal{I}_{-\mathbf{s}_*}} \Delta_{\mathbf{s}} \sum_{t \in \mathcal{R}} \mathbb{1}\{\overline{\mathcal{S}}_t = \mathbf{s}, E_t, \mathcal{P} \in \mathcal{C}_t\}$$

$$+ \mathbb{E}\left[\sum_{t=1}^{T} \mathbb{1}\{\underline{\mathbf{S}}_t\}\right] + \sum_{t \in \mathcal{R}} \mathbb{P}(E_t^{\mathbf{C}}) + \sum_{t=1}^{T} \mathbb{P}(\mathcal{P} \notin \mathcal{C}_t),$$

where the first three terms present regret from exploitation, exploration with greedy approximation, and forced exploration respectively. So the proof can be concluded the proof by substituting (20), (25), (23), (28) and (31) into the above inequality. $\qquad\square$

## C  Experiment Details and Additional Experimental Results

In the main paper, we conducted experiments in three causal bandit instances. The causal graph corresponding to each instance is shown in Fig. 2. For $p \in [0,1]$, let $\mathcal{B}(p)$ denote a Bernoulli random variable with probability $p$ to be 1 and $1 - p$ to be 0. The detailed structural equations for each causal bandit instance are described below.

**Task1:** $U_{V_1} = \mathcal{B}(0.6)$, $U_{V_2} = \mathcal{B}(0.11)$, $U_{V_2 V_3} = \mathcal{B}(0.51)$ and $U_{V_3} = \mathcal{B}(0.15)$.

$$f_{V_1}(u_{V_1}) = u_{V_1}$$
$$f_{V_2}(v_1, u_{V_2}, u_{V_2 V_3}) = v_1 \oplus u_{V_2} \oplus u_{V_2 V_3}$$
$$f_{V_3}(v_2, u_{V_3}, u_{V_2 V_3}) = v_2 \oplus u_{V_3} \oplus u_{V_2 V_3} \oplus 1$$

**Task2:** $U_{V_1} = \mathcal{B}(0.45)$, $U_{V_2} = \mathcal{B}(0.05)$, $U_{V_5 V_2} = \mathcal{B}(0.54)$, $U_{V_3} = \mathcal{B}(0.07)$, $U_{V_3 V_4} = \mathcal{B}(0.51)$, $U_{V_4} = \mathcal{B}(0.06)$ and $U_{V_5} = \mathcal{B}(0.06)$.

$$f_{V_1}(u_{V_1}) = u_{V_1}$$
$$f_{V_2}(u_{V_2}, u_{V_5 V_2}) = u_{V_2} \oplus u_{V_5 V_2}$$
$$f_{V_3}(v_1, u_{V_3}, u_{V_3 V_4}) = v_1 \oplus u_{V_3} \oplus u_{V_3 V_4}$$
$$f_{V_4}(v_2, u_{V_4}, u_{V_3 V_4}) = 1 \oplus v_2 \oplus u_{V_4} \oplus u_{V_3 V_4}$$
$$f_{V_5}(v_3, v_4, u_{V_5}, u_{V_5 V_2}) = v_3 \oplus v_4 \oplus u_{V_5} \oplus u_{V_5 V_2}$$

**Task3:** $U_{V_1} = \mathcal{B}(0.5)$, $U_{V_2} = \mathcal{B}(0.85)$, $U_{V_3} = \mathcal{B}(0.85)$, $U_{V_3 V_7} = \mathcal{B}(0.85)$, $U_{V_4} = \mathcal{B}(0.14)$, $U_{V_5} = \mathcal{B}(0.74)$, $U_{V_6} = \mathcal{B}(0.74)$, $U_{V_7} = \mathcal{B}(0.54)$.

$$f_{V_1}(u_{V_1}) = u_{V_1}$$
$$f_{V_2}(u_{V_2}) = u_{V_2}$$
$$f_{V_3}(v_1, u_{V_3}, u_{V_3 V_7}) = (v_1 \oplus u_{u_{V_3}}) \vee u_{u_{V_3 V_7}}$$
$$f_{V_4}(v_2, u_{V_4}) = v_2 \wedge u_{V_4}$$
$$f_{V_5}(v_3, v_4, u_{V_5}) = (v_3 \oplus u_{V_5}) \vee v_4$$
$$f_{V_6}(v_4, u_{V_6}) = v_4 \vee u_{V_6}$$
$$f_{V_7}(v_5, v_6, u_{V_5}, u_{V_7}, u_{V_3 V_7}) = ((v_5 \oplus v_6 \oplus 0 \oplus u_{V_3 V_7}) \wedge u_{V_7}) \oplus v_6$$

