# OpenReview forum: "Approximate Allocation Matching for Structural Causal Bandits with Unobserved Confounders"
_NeurIPS.cc/2023/Conference — NeurIPS 2023 poster_

### Official Review · Reviewer_EzBh · 2023-06-21

**Soundness:** 4 excellent
**Presentation:** 2 fair
**Contribution:** 3 good
**Rating:** 7
**Confidence:** 3

**Summary:**

The paper presents two primary contributions for the causal bandit problem with unobserved confounders, general graphs, and discrete RVs.
1) a lower bound on the regret any algorithm can obtain for this problem.
2) an upper bound on the regret of a newly proposed algorithm that is able to generalize across interventions (unlike an algorithm such as vanilla UCB) using allocation matching.

**Strengths:**

- This is the first causal bandit algorithm I've seen that has a regret guaratnee for the cumulative regret setting and general graphs (besides [20], which focuses only on reducing the number of arms rather than using structure to generalize across arms). I think the community will find this very interesting.
- Generally well written prose.
- There is not much to add regarding the strengths: the paper is generally clearly presented and the result is a novel solution to a open problem that the community has shown a good amount of interest in. I have not gone through the supplementary material so I cannot comment on the correctness of the proofs.

**Weaknesses:**

- Experiments are synthetic (fairly common in these causal bandit papers in fairness) but also on extremely small graphs. It would be good to understand how performance varies with graph size and the size of the intervention set.
- It would be nice to see a discussion in words of how the regret bound scales with various quantities- graph size, intervention set size. The bound as presented is not one of the easiest in terms of extracting this kind of qualitative information.
- Some prose was clunky and I recall a few typos. The text should be run through Grammarly or a similar tool.

**Questions:**

- Line 293 onwards compares the regret guarantee to [20]. Is it possible to state exactly that the regret bound of your approach is upper-bounded by theirs? I couldn't directly infer this from the comparison you gave.
- The algorithm seems to be anytime (doesn't need to know T) but would be great to have this clarified explicitly.
- Also seems like you can have arbitrary constraints on the interventions and also that multi-target interventions can be used in your algorithm. Is this correct?
- How computationally efficient is the algorithm? In the computation of the UCB (in algo 1 and remark 2) it looks like you sum over all possible configurations of the variable in the graph, which will explode as the number of nodes gets larger. The graphs in the experiments are also very simple/small so I'm wondering how scalable this method is.
- Can you clarify what and how many additional hyperparameters the method requires compared to UCB? How do you set these hyperparameters in the experiments of the main paper? Look like you additionally have just $\epsilon$ and $\gamma$.
- Standard UCB can be used without a forced exploration step- why/how does your algorithm break without it? I'm more familiar with regret guarantees that are independent of $\Delta_s$ (so where regret is on order of sqrt(T)). What happens here as $\Delta_s$ gets very small or very large?  The forced exploration likely explains why for small time horizons the proposed algorithm is not as good as the baselines on some graphs.

**Limitations:**

- In the related work the nonparametric setting studied here is presented as a benefit but it should also be mentioned that the fully discrete setting here can be limiting compared to works that study continuous observations (it is also not really clear that the main ideas described here can be easily extended to those settings).
- The computational intensity of the algorithm, especially depending on graph size and intervention set size, should be discussed.
- Line 338 "it is shown that SCM-AAM has superior empirical performance compared to existing baseline algorithms". I think this is potentially misleading since the baseline compared to is from one of the earliest casual bandit papers and likely not the strongest among published baselines on the graphs used. I will not insist on comparing against every or any other causal bandit algorithm published since then but I think the claim could be made more moderate, rather than implying that some kind of SOTA empirical performance has been shown.

---

> ### Author Rebuttal · Authors · 2023-08-08
>
> Thank you for your valuable feedback and supportive comments. Your questions will help us improve the clarity and presentation of the paper.  Please see our responses below.
>
> “size of graphs in experiments”: We agree with the reviewer that extending our simulations to larger graphs will help demonstrate the efficacy of our algorithm further. In the camera-ready version, we will test the proposed algorithm on larger graphs and discuss how performance varies with graph quantities, which could potentially include the number of nodes, size of the intervention set, and connectivity.
>
> "regret bound vs graph quantities": Due to the complicated space of causal models characterized by the canonical SCM formulation, it is hard to relate the regret bounds to graph quantities. However, our algorithm is designed with the purpose to match the abstract regret lower bound. We also believe the regret bounds have a sublinear relationship with the size of the intervention set since interventions are not treated as independent actions. By applying the canonical SCM model, the dependencies of interventional distributions are exploited to reduce regret.
>
> "compare the regret guarantee to [20]": Disregarding the constants $\gamma$, $\kappa$, and $\epsilon$, the regret bound of the proposed algorithm is smaller than that of KL-UCB + POMIS [20]. The reason is that our UCB index is computed from a more constrained space compared with KL-UCB. Please also notice that the constants $\gamma$, $\kappa$, and $\epsilon$ can be arbitrarily small. In the camera-ready version, we will revise line 293 to line 303 to clarify this point.
>
> "is the algorithm anytime": Yes, the algorithm is anytime. We will clarify this in the camera-ready version.
>
> "arbitrary constraints on the interventions": Multi-target interventions are used in our algorithm and arbitrary constraints can be put on interventions. However, we chose to apply interventions on Possible Optimal Minimal Sets POMIS (POMIS) to reduce the action space. We will clarify this in section 5.
>
> "computation efficiency of the algorithm": We report the experimental computation time in Appendix Section C, though the graphs are small. The runtime of SCM-AAM is longer compared with others since it processes more information. The expression in algorithm 1 and remark 2 enumerates all configurations of variables for ease of notation. The algorithm only needs to deal with variables related to the reward node. By only intervening on POMIS, these variables are contained in Unobserved-Confounders’ Territory defined in [20], which can still be many for certain causal graphs. However, in applications where decisions can take time, e.g. policy-making in companies, the proposed algorithm would be valuable. Besides, we believe there exists a trade-off solution with low computation complexity but higher regret which we would like to explore in future extensions of this work.
>
> We will add additional simulation results on larger graphs and make a detailed discussion about how graph size and intervention set size affect the computational intensity of the algorithm.
>
> "additional hyperparameters": Please note that since our algorithm leverages the structured bandit framework, it requires two parameters $\epsilon$ and $\gamma$. The $\epsilon$ corresponds to forced exploration and $\gamma$ provides a buffer for the confidence region. They will affect short-term performance and the consistency of SCM-AAM.  With a long horizon length, we suggest selecting them to be small. In our experiments, we did not explicitly tune these hyperparameters. Our results show that even without fine-tuning, SCM-AAM still has reasonably good performance.
>
> “why forced exploration?”: To leverage the causal graph to exploit the dependencies among different interventions, our algorithm requires each interventional distribution to converge to the ground truth at a fast enough rate. To achieve it, forced exploration allocates a fair amount of exploration to each intervention. Otherwise, the regret cannot be approximately bounded by $\sum_{s \in \mathcal{I}_{-*}} \Delta_s \zeta_s(\mathcal{P})  \ln T$ in Theorem 2, where $\zeta_s(\mathcal{P})$ is a factor depending on true interventional distributions in $\mathcal{P}$.
>
> Note that the poor short-term performance is typical for allocation matching-based algorithms, which we agree could be due to forced exploration.
>
> "logarithmic regret vs $O(\sqrt{T})$ regret": Regret on order of $\sqrt{T}$ corresponds to the works-case regret for a given horizon $T$, as mentioned in JY. Audibert, and S. Bubeck (2009). Say, $\Delta_s$ is close to $1/\sqrt{T}$, it is hard to differentiate the optimal action from the suboptimal ones, so the regret will be in the order of $\sqrt{T}$. If $\Delta_s$ for each intervention $s$ is much greater than $1/\sqrt{T}$,  the logarithmic regret better characterizes the performance.
>
> “discrete vs. continuous settings”: We agree that the fully discrete setting in this paper has its limitations and can not be easily extended to the continuous observation setups.
>
> For minor comments:
>
> We will revise Line 338 into “SCM-AAM exhibits promising performance in comparison with classic baseline algorithms.” Also, we will carefully check the writing and improve the presentation in the revised draft.
>
> We would be happy to answer any further questions the reviewer might have. Thank you once again!
>
> $Reference$
>
> JY. Audibert, and S. Bubeck. "Minimax Policies for Adversarial and Stochastic Bandits." COLT. Vol. 7. 2009.

---

> > ### Comment · Reviewer_EzBh · 2023-08-15
> > **Re**
> >
> > Thank you to the authors for their reply. I am writing to confirm that I've read the author's rebuttal and my scores remain unchanged.

---

### Official Review · Reviewer_v6qA · 2023-07-04

**Soundness:** 3 good
**Presentation:** 2 fair
**Contribution:** 3 good
**Rating:** 5
**Confidence:** 3

**Summary:**

This paper studies the structural causal bandits in which the causal graph contains latent confounders. The paper contains a lower-bound on attainable regret as well as an algorithm that achieves instance-dependent upper bound.

**Strengths:**

The paper studies an established problem, building primarily on [20] and the idea of intervening on POMIS. The paper develops a number of theoretical results on attainable regret.


**Weaknesses:**

The paper is quite dense and could benefit from a more clear exposition on its theoretical contributions. For instance, this may include close-form or approximate values on the various factors listed in L228, L291.

To underscore the advantage of SCM-AAM and to establish novelty, it seems that one has to establish SCM-AAM is better than the algorithm developed in [20]. The comparison part L293-303 (particularly L298-L301) could benefit from greater clarity on the precise benefit of this paper's algorithm over that of [20].  It would be nice to see a particular factor, say in a particular causal bandit instance where it is clear the algorithm attains better regret. It seems like graphs in Figure 2 are promising candidates, as it is shown empirically that SCM-AAM does better.

**Questions:**

Please see above.

**Limitations:**

Seems fine.

---

> ### Author Rebuttal · Authors · 2023-08-08
>
> Thank you for your valuable feedback. Please see our responses below.
>
> In L228 and L291, the upper and lower bounds depend on the topology of the causal graph. The complicated space of general causal models is not amenable to a closed-form result or an approximated value. Instead, the factors appearing in L228 and L291 are represented as the value of the presented optimization problems. We will add this point in the conclusion as a limitation of our work.
>
> In [20], applying intervention only on POMISs can help reduce the set of candidate actions at the beginning of a causal bandit algorithm. However, any such purely topology-based approach would treat each “arm” independently during the algorithm execution. Whereas our structured bandit-based method can incorporate the sampling results from these arms together during the execution of the algorithm as new samples are collected by leveraging the canonical SCM-based idea.
>
> In L293-303, we compare the regret upper bound of the SCM-AAM and [20] to provide theoretical support for the SCM-AAM performing better. In the experiment section, we show SCM-AAM has better empirical performance in 3 different tasks. SCM-AAM can leverage dependencies in the interventional distributions to reduce regret.
>
> We hope that our responses will address the reviewer's concerns about this paper and increase the overall score. We are also happy to discuss any further questions. Thanks.

---

### Official Review · Reviewer_BoCS · 2023-07-05

**Soundness:** 3 good
**Presentation:** 2 fair
**Contribution:** 1 poor
**Rating:** 4
**Confidence:** 4

**Summary:**

The authors present a method for studying structural causal bandits with unobserved confounders. They show that their method, which unlike the baseline (POMIS), includes two tuning parameters, has superior empirical performance on three synthetic experiments (three synthetic SCMs).

**Strengths:**

The main review can be found in the Questions section.

**Weaknesses:**

The main review can be found in the Questions section.

**Questions:**

## Abstract

- Bandits do not have to be used online, offline usage if common as well. Please consider revising the first line.
- An SCM does not model random variables as such; it specifies a distribution over the exogenous variables in the model (the noise variables) - but the endogenous variables are not random but are specified by the design and typically user experience.
- Perhaps it would be better to say that the reward variable (everyone uses Y) returns the stochastic reward instead of the non-manipulable variables, of which there can be many but causal bandit only has one reward variable.
- It is called the ‘exploration-exploitation’ trade-off, please fix this in your abstract.
- I would urge the authors to considerably shorten their abstract. It is meant to short summary of your method and main results. I fear that at present length it is eating into valuable space in the paper, space which could be better spent on explaining your method.

## Introduction

- I think a bit more precision is in order in this section, you say on line 54, that “it [causal inference] has special value”. This is true, but perhaps it would be better to explicate on what that value actually constitutes and the utility it has compared alongside purely associative statistics (correlation-based statistics).
- Lines 63-70 require _a lot_ of background knowledge to be read with any sort of utility. The POMIs, for example, requires an entire paper (Lee and Barenboim) to expose properly or you could just state the main result. But you haven’t instead you had merely mentioned POMIS but without telling the reader what it is order what it does. That is not very helpful particularly since the POMIS is such a powerful concept. Equally you have no adequately exposed the KL-UCB algorithm beyond stating it with its name. That does not do it justice and certainly not in the causal bandit context.
- On line 69 you say: “A general linear model is adopted in [12], which is different from the non-parametric setup in this paper”  - but _how_ is it different? What is the non-parametric setup in this paper? What is a linear model in this context? You need to be much more precise in order for your readers to follow your line of reasoning.
- On line 72: how can the arms be related to an arbitrary causal diagram? They are functions of the topology and that in itself is a function of the structural equation models specified in the SCM (under Pearl’s models of causation). The _domains_ of the arms may be arbitrary (though that would be rather odd). Hence, I am not quite what you mean by this sentence. Please do explain.

## Problem Statement and Background

- I am not sure you even need to state assumption 1, it is such a common assumption with causal bandits (indeed most structural bandits) that it is left unsaid and assumed given.
- I think you mean to say on line 132: “means the expectation is computed with [respect to the] probability distribution P”.
- Is there a specific reason you name your reward variable $V_n$?
- Line 150: what are you actually referencing with [36]? Your reference is an entire book but what more precisely are you giving reference to?
- With assumption two, are you suggesting that there are instances where the POMIS does not contain the optimal set or two or more sets in the POMIS have the same reward? You suggest in the line above the assumption that certain SCMs do not necessarily obey this assumption? Could you please explain further what you mean as this seems to contravene the results by Lee and Barenboim’s work on intervention sets.

## Response Variables and Space of Interventional Distribution Tuples

- Please explain how $V_2$ is a c-component, I do not follow; it is not connected to any bi-directed edges in figure 1(a)

## SCM-based Causal Bandit Algorithm

- SCM-AAM has two tuning parameters compared to the more canonical POMIS selection (merely a function of the topology) - it seems like you are making things more complex by introducing tuning parameters rather than just leveraging the topology of the DAG, as is the case with the MIS and POMIS. Can you please comment on this.
- I really like how this section is exposed and told. Good job.

## Finite time regret analysis

- Please comment on settings where G does not contain any confounders (dashed edges). Is there any utility by using SCM-AAM then? Or do the results then fall back to those in reference [20]?

## Experimental results

- I think it would be more fair for you to use the Thompson sampling and KL-UCB algorithms instead of vanilla UCB without structural information, that does not sound like a reasonable choice in this setting. Or even the algorithms used by Lattimore and Lattimore in their 2016 paper.
- Why do you choose those values for your hyperparameters gamma and epsilon? Have you conducted an ablation study for different variable settings?
- The shape of the purple curve is not particularly surprising since it does not have access to causal information. As noted; this is not a fair comparison and you should leverage other, fairer methods, for comparison just to make it an even ‘fight’. There is very little to learn from the purple curve given that Lee and Barenboim have already shown that algorithms with causal information perform better than vanilla algorithms (Lattimore and Lattimore also commented on this idea).

**Limitations:**

The main review can be found in the Questions section.

I think the title is a bit misleading; plenty of studies have considered causal bandits with unobserved confounders (Lee and Bareinboim being a pair of authors who have published a number of works in this area). Why do you draw attention to this aspect of your method? Would be interested to hear your thoughts on this.

---

> ### Author Rebuttal · Authors · 2023-08-08
>
> Thank you for your valuable feedback and constructive comments. We hope that our responses will address the reviewer's concerns.
>
> "title is a bit misleading": Please note that we are not claiming that ours is the first causal bandit algorithm with unobserved confounders. Our goal with the title was to explain that our setting contains unobserved confounders. The emphasis is on the approximate allocation matching solution using structural bandits.
>
> $Abstract: $
>
> We will shrink the abstract to only deliver the key information about the paper. Besides, we will remove the word ‘online’ and use ‘exploration-exploitation’ instead of ‘explore-versus-exploit’. To be clearer about the reward variable, we mention “the reward variable that is non-manipulable returns the stochastic reward”.
>
> $Introduction:$
>
> Thank you for your insightful comment. We will revise line 50-54 as follows: Compared to the classic MAB, casual bandit represents a more detailed model of real-world decision-making problems with richer feedback from multiple random variables, and different actions are interrelated in a causal manner. In applications such as social and economic experiments where the space of actions is large and explicit experimentations are costly, applying causal reasoning gives us the benefit of exploiting the dependecies between actions, so that the learning process can be accelerated and more rewarding interventions can be selected.
>
> We apologize for not providing sufficient detail about POMIS due to space constraints. We will make sure to add details of the POMIS idea, and KL-UCB with the additional page provided in our camera-ready version.
>
> By non-parametric we mean no assumptions on the functions of the SCM, whereas in a linear model, the functions are assumed to be linear. We will clarify.
>
> We agree the current line 72 is imprecise. We mean there is no constraint on the causal graph in the causal bandit problem. For an arbitrary causal bandit with confounders, the theory fits and the algorithm can be applied. We will revise accordingly.
>
> $Problem$ $Statement$ $and$ $Background:$
>
> We added Assumption 1 to clearly distinguish our setting from some other bandit settings where one might have uncountably many arms such as linear bandits in Y. Abbasi-Yadkori et.al. (2011).
>
> “naming of reward node $V_n$”: This was just a convention. We will clarify this.
>
> We cite [36] for the concept of general position assumption. We will point to the exact page and theorem.
>
> Assumption 2 does not contradict with Lee and Bareinboim (2018). Our emphasis was not on the existence but on the uniqueness of the optimum. Yes, the existence is implied by Lee and Bareinboim (2018) but not uniqueness. Note that multiple sets in POMIS can potentially attain the optimum.
>
> "$V_2$ is a c-component in figure 1(a)": Please note that a singleton node without bidirected edges is, by definition of c-component in Tian and Pearl (2002), a c-component. We will revise the draft to clarify this.
>
> $SCM-based$ $Causal$ $Bandit$ $Algorithm$:
>
> "tuning parameters in the algorithm": We agree that POMIS can help reduce the set of candidate actions at the beginning of a causal bandit algorithm. However, please observe that any such purely topology-based approach would treat each “arm” independently during the algorithm execution. Whereas our structured bandit-based method can incorporate the rewards from these arms together during the execution of the algorithm as new samples are collected by leveraging the canonical SCM-based idea.
>
> Accordingly, the SCM-AAM algorithm employs the allocation matching technique, which requires tuning parameters $\epsilon$ and $\gamma$ regarding forced exploration and size of confidence region (see equation (7)).  With a long horizon length, we suggest selecting them to be small. In our experiments, we did not explicitly tune these hyperparameters. Our results show that even without fine-tuning, we outperform existing works.
>
>
> $Finite$ $time$ $regret$ $analysis$:
>
> "settings where $G$ does not contain any confounders": As indicated by [20], when G does not contain any confounders, the POMIS will be the parents of the reward node. In this case, the result falls back to those in [20]. This is a great point and we will add this as a remark.
>
> $Experimental$ $results$:
>
> Please note that Lattimore and Lattimore (2016) is in the pure exploration setting. Thus, we expect it to perform badly for cumulative regret, which is our metric. If the reviewer feels strongly about this comparison, we can add it in camera ready.
>
> We agree with the reviewer and we will remove the purple line.
>
> We omitted KL-UCB because we are already comparing against KL-UCB with POMIS [20], which we expect to perform much better.
> In the camera-ready version, we will add Thompson sampling (TS) with POMIS experimental results, which are shown in the PDF file attached to author rebuttal. We show Thompson sampling on POMIS performs better than KL-UCB on POMIS, while our method outperforms both.
>
> We hope that our responses will address the reviewer's concerns and increase the overall score. We are also happy to discuss any further questions. Thanks.
>
> $References$
>
> Y. Abbasi-Yadkori, Yasin, D. Pál, and C. Szepesvári. "Improved algorithms for linear stochastic bandits." Advances in neural information processing systems 24 (2011).
>
> J. Tian and J. Pearl. "A general identification condition for causal effects." Aaai/iaai. (2002).
>
> F. Lattimore, T. Lattimore, and MD. Reid. "Causal bandits: Learning good interventions via causal inference." Advances in neural information processing systems 29 (2016).

---

> > ### Comment · Reviewer_BoCS · 2023-08-16
> >
> > I thank the authors for taking the time to respond to my comments. I still believe the papers warrants major revision to address many of my points but I am confident the authors can address in the next version.

---

> > > ### Author Response · Authors · 2023-08-16
> > >
> > > Thank you very much for the follow-up comment. We appreciate the reviewer's confidence in us and would like to reiterate our gratitude for their feedback.
> > >
> > > However, we respectfully disagree with the assessment that implementing these changes requires a major revision. We will definitely more clearly explain the POMIS idea and elaborate on the value of a causal approach, and the value of utilizing the structure. If the reviewer still believes that our paper requires a major revision, we would be very happy to hear the reasons to be able to better address them in our revision.

---

> > > > ### Comment · Area_Chair_DQWm · 2023-08-21
> > > > **To Reviewer BoCS**
> > > >
> > > > Dear Reviewer BoCS,
> > > >
> > > > In your last message, you mentioned that you still believe the paper needs a major revision to address many of your comments, while the authors believe most of the comments can be addressed relatively easily. Can you substantiate the main technical issues that you believe warrant a major revision? Thanks.
> > > >
> > > > Area Chair

---

### Official Review · Reviewer_gxS9 · 2023-07-23

**Soundness:** 3 good
**Presentation:** 3 good
**Contribution:** 3 good
**Rating:** 6
**Confidence:** 2

**Summary:**

This paper considers online learning in a structural causal bandit whose stochastic environment is govened by causal relations represented by an SCM. The paper gives a logarithmic asymptotic regret lower bound characterized by an optimization problem, and inspires on that, provides an algorithm to untilize the causal structure and accelerate learning process. Numerical experiments illustrate that the proposed algorithm outperforms traditional online learning bandit algorithms.

**Strengths:**

1. Writing. The paper has a clear organization of the relative background, concerning Structural Causal Model, Causal Bandit, and Interventional Distribution Tuples in Section 2-3. The presentation is good and easy to follow.
2. Originality. The paper studies online decision-making algorithms for causal bandits. The lower bound result extends from similar results in the structured bandit [1], and so as corresponding algorithm.
3. Quality. The writing is of high quality, and theoretical results and experiments are enough to support the algorithm.

[1] R. Combes, S. Magureanu, and A. Proutiere. Minimal exploration in structured stochastic bandits. Advances in Neural Information Processing Systems, 30, 2017.

**Weaknesses:**

1. The whole paper seems to extend the methodology and algorithm in [1] to causal bandit settings. It is better if the author could highlight extra contributions of this paper.

[1] R. Combes, S. Magureanu, and A. Proutiere. Minimal exploration in structured stochastic bandits. Advances in Neural Information Processing Systems, 30, 2017.

**Questions:**

See above.

**Limitations:**

See above.

---

> ### Author Rebuttal · Authors · 2023-08-08
>
> We first thank the reviewer for valuable feedback and supportive comments.  Please see below for our response.
>
> “highlight extra contributions of this paper”:
>
> The major contribution of this paper is utilizing a canonical SCM formulation, which partitions the space of causal models, to characterize a parametric space of interventional distributions compatible with a causal graph. Leveraging such a space in the setting of approach in [1], our algorithm is able to effectively utilize sampling history to reduce uncertainty. Note that this connection and leveraging of this canonical SCM formulation in this context, to the best of our knowledge, is new.
>
> Additionally, in our algorithm, we propose to convert the non-convex infinite program in [1] to a convex program for the causal bandit problem. And due to this new algorithm design, we provide a revised analysis of regret upper bound. In the proof, we give a detailed analysis of the continuity property of the UCB index (related to the canonical SCM formulation) with respect to interventional distributions. And based on it, the expected number of times each suboptimal intervention is selected is bounded.
>
> We would be happy to answer any further questions the reviewer might have. Thank you once again!

---

### Author Rebuttal · Authors · 2023-08-08

We thank the reviewers for their valuable feedback and AC for the assistance during the review process. We address each review’s comments in separate rebuttals. In response to reviewer BoCS, the attached PDF file includes additional experiment results of Thompson Sampling (TS) in comparison with the proposed algorithm in this paper. We observe that TS on POMISs performs better than KL-UCB on POMIS while our method outperforms both.

---

### Decision · Program_Chairs · 2023-09-21

**Decision:**

Accept (poster)

**Comment:**

The paper studies the structural causal bandit with unobserved confounders. Reviewers are in general positive to the paper, acknowledging that the paper studies an important problem with novel solutions, and it will benefit the research community. The concerns are mostly on the writing of the paper, and the authors mostly address these concerns. I believe a careful revision by the authors will address all the presentational issues raised by the reviewers. Therefore, I would like to recommend acceptance to the paper, while encouraging the authors to do a thorough revision on the paper addressing all the concerns from all the reviewers.